# Learning to Generate Visual Questions
# with Noisy Supervision

**Kai Shen**[*†], **Lingfei Wu**[*‡], **Siliang Tang**[§†], **Yueting Zhuang**[†],
**Zhen He**[‡], **Zhuoye Ding**[‡], **Yun Xiao**[‡], **and Bo Long**[‡]

[†]Zhejiang University    [‡]JD.COM

shenkai@zju.edu.cn, lwu@email.wm.edu, {siliang,yzhuang}@zju.edu.cn,
{bjhezhen,dingzhuoye,xiaoyun1,bo.long}@jd.com

## Abstract

The task of visual question generation (VQG) aims to generate human-like neural questions from an image and potentially other side information (e.g., answer type or the answer itself). Existing works often suffer from the severe one image to many questions mapping problem, which generates uninformative and non-referential questions. Recent work has demonstrated that by leveraging double visual and answer hints, a model can faithfully generate much better quality questions. However, visual hints are not available naturally. Despite they proposed a simple rule-based similarity matching method to obtain candidate visual hints, they could be very noisy practically and thus restrict the quality of generated questions. In this paper, we present a novel learning approach for double-hints based VQG, which can be cast as a weakly supervised learning problem with noises. The key rationale is that the salient visual regions of interest can be viewed as a constraint to improve the generation procedure for producing high-quality questions. As a result, given the predicted salient visual regions of interest, we can focus on estimating the probability of being ground-truth questions, which in turn implicitly measures the quality of predicted visual hints. Experimental results on two benchmark datasets show that our proposed method outperforms the state-of-the-art approaches by a large margin on a variety of metrics, including both automatic machine metrics and human evaluation.

## 1 Introduction

Recent years have witnessed a surge of interest in the Visual Question Generation (VQG) task. VQG aims to generate human-like neural questions from an image and potentially other side information (e.g., answer type or the answer itself). VQG is one of the effective data augmentation approaches for providing high-quality synthetic training data for visual question answering (VQA) [28] and visual dialog system [19]. Moreover, VQG models are also particularly useful for the few-shot learning or zero-shot learning [36, 44]. Conceptually, VQG is a very challenging task since the generated questions are not only required to be consistent with the image content but also meaningful and answerable to humans.

To handle the VQG task, recent approaches explore three directions mainly. One direction is focusing on VQG without any hints. Although more diverse questions can be generated, it often leads to non-informative questions and hard to find any corresponding answers. Furthermore, in view of data augmentation, a successful VQG system should be goal-driven in order to provide valuable additional QA pairs to further enhance generalization capability of its dual VQA task [22].

---

[*]Both Authors Contributed Equally. This work was conducted when the first author was an intern in JD.COM.
[§]Corresponding Author

35th Conference on Neural Information Processing Systems (NeurIPS 2021).

Another direction utilizes answer-type hints for VQG in order to generate more referential questions while exploiting diverse questions. However, given one image, there are often multiple questions with the same question types (i.e., what, how, and how many). If only answer-type is used, the generated questions are often either not referred to any existing answers or not informative questions. As a result, these generated questions either cannot be used for training VQA or can be too noisy to degenerate the generalizability of VQA.

The last direction exploits the textual answer information as answer hints for VQG. This is particularly useful when two questions are the same question types but different questions for an image. As a result, the answer hint can often greatly reduce the ambiguities when learning to generate visual questions. Therefore, it can train a better question generator for more diverse questions instead of specific questions. Unfortunately, there are still many cases where answer hints are not informative enough to reduce the ambiguities for generating a question, especially when there are multiple visual objects that are associated with answer hint.

Recently, Kai et al. [20] presented a novel double-hints approach for VQG, which showed that the model can faithfully generate much better quality questions by leveraging double visual and answer hints. However, visual hints are not available naturally, which requires highly expensive human annotations. To handle this issue, they proposed a simple rule-based similarity matching method to obtain a set of visual hints, which are often too noisy to generate high-quality questions. As a result, the performance of VQG could be highly restricted by these low-quality visual hints.

Therefore, an important question is, without any expensive human annotation, can we identify important salient visual regions of interest associated with answer hints to overcome the above deficiencies? In this paper, we present a novel learning approach for double-hints based VQG, which can be cast as a weakly supervised learning problem with noises. The key rationale is that, the salient visual regions of interest can be viewed as a constraint to improve the generation procedure for producing high-quality questions. As a result, given the predicted salient visual regions of interest, we can focus on estimating the probability of being ground-truth questions, which in turn implicitly measures the quality of predicted visual hints.

To this end, we propose a Double-Hints guided Generative Adversarial Networks (DH-GAN) consisting of a double-hints based question generator and a question-answer-aware discriminator. In particular, our generator has a visual hints predictor with a cross-modal reasoning module to determine the salient visual regions associated to the corresponding question, and a question generation module with predicted visual hints and textual answer hints. Our discriminator is to distinguish whether a sample triplet (i.e., the image, answer, and question) is generated from the generator or ground-truth. Moreover, we design a novel hybrid reward function that combines the generator and discriminator so that it encourages the model to perform both exploration of question sample and the visual hints via policy gradient. Experimental results on two benchmark datasets show that our proposed DH-GAN outperforms the state-of-the-art approaches by a large margin on a variety of metrics, including both automatic machine metrics and human evaluation.

## 2 Double-hints Guided Generative Adversarial Networks for VQG

In this section, we first introduce the double-hints based visual question generation (VQG) problem and then discuss the overall model architecture. Next, we detail each key components of our proposed DH-GAN model for double-hints based VQG in the subsequent sections.

### 2.1 Problem Formulation and Overall Model Architecture

**Problem Formulation.**   Conceptually, the goal of VQG is to generate natural questions based on a given form of visual data (i.e., images [22] and videos [46]), together with other side information (i.e., answers [28], answer-types [22], or both textual answers and visual regions [20]). In this paper, we focus on the task of generating high-quality questions from images with double text and visual hints.

Given an image $I$, we assume an image is represented by a set of visual regions $\mathcal{R} = \{r_1, r_2, ..., r_N\}$ with initial embedding $\mathbf{V} = [\mathbf{v}_1, \mathbf{v}_2, ..., \mathbf{v}_N]$ via an object detection method [40], where $N$ is the number of visual regions. Similarly, we assume the answer $A = \{a_1, a_2, ..., a_m\}$ with initial embedding $\{\mathbf{a}_1, \mathbf{a}_2, ..., \mathbf{a}_m\}$ and the target question $Q = \{q_1, q_2, ..., q_n\}$ with initial embedding $\{\mathbf{q}_1, \mathbf{q}_2, ..., \mathbf{q}_n\}$ are a collection of words, respectively. Next, the visual hints $\mathcal{V} = \{r_1, r_2, ..., r_T\}$

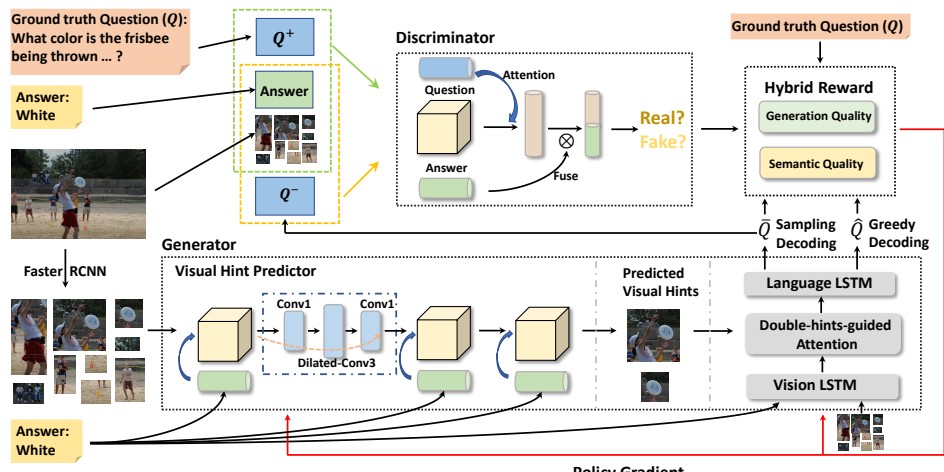

Figure 1: The proposed DH-GAN Model Architecture with Noisy Supervision.

are the visual regions of interest, which are (semantically) associated with the corresponding textual answer. Note that, $\mathcal{V}$ is the subset of $\mathcal{R}$ ($\mathcal{V} \subseteq \mathcal{R}$). The double-hints based VQG task aims to generate not only answerable but also referential questions based on the given image $I$ and the double hints $A$ and $\mathcal{V}$, which maximizes the following conditional likelihood

$$\hat{Q} = \text{argmax}_Q P(Q|I, A) = \text{argmax}_Q [P(\mathcal{V}|I, A)P(Q|I, A, \mathcal{V})] \tag{1}$$

where the visual hints $\mathcal{V}$ are obtained through the alignments between the image $I$ and the answer $A$. Note that, since there are no additional human annotations when obtaining target visual hints during pre-processing, it is not surprising that they are very noisy, which leads to the non-accurate prediction of $\mathcal{V}$ during inference.

**The proposed DH-GAN Framework.** Despite the potentially useful additional information the visual hints may carry, how to identify the salient visual regions of interest from a set of visual region candidates without any human annotations is a very challenging task. Therefore, it is crucial that our proposed model is able to learn to generate visual questions with noisy supervision. In order to effectively address this challenge, we propose the DH-GAN model, which consists of two key components: a generative model $G$ and a discriminative model $D$ as shown in Fig. 1.

Formally, we denote the sample triplet as $x = \{I, Q, A\}$ ($I$ for image, $Q$ for question, and $A$ for answer). We further denote $x^+ = \{I, Q^+, A\}$ as the human-annotated sample ("real") and $x^- = \{I, Q^-, A\}$ as the sample generated by the system ("fake"). Generally speaking, the goal of the generator $G$ is to first select the salient visual hints from the input image and the textual answer and then generate the question based on the predicted visual hints and the corresponding textual answers. Since there is no direct alignment information between the visual hints and answer, the goal of the discriminator $D$ is to implicitly measure the quality of predicted visual hints by judging whether the current sample triplet is "real" or "fake". In other words, the generator $G$ and discriminator $D$ undergo an adversarial process to encourage the generator to identify the salient visual region of interest so that the high-quality question can be generated, which are indistinguishable from the "real" ones by the discriminator $D$. Once the Nash Equilibrium is achieved via GAN, the generated question $Q^-$ would be very close to $Q^+$ in the discriminator's embedding space semantically. In what follows, we will discuss each component of the proposed DH-GAN in detail.

## 2.2 G-step: Double-hints Guided Question Generator

Since visual hints are not available naturally, it is unclear how to effectively identify the salient visual hints and leverage them for generating the high-quality question. In this section, we will introduce two important steps of G-step: i) identifying visual hints and ii) generating questions with double visual and text hints.

### 2.2.1 Identifying Visual Hints

To exploit the rich correlations of the visual regions, inspired by ResNet [17], we utilize the following relation-aware encoder module:

$$\mathbf{V}_{out} = Conv_1(DilatedConv_3(Conv_1(\mathbf{V})) + \mathbf{V} \tag{2}$$

where $\mathbf{V} \in \mathbb{R}^{N \times d}, \mathbf{V}_{out} \in \mathbb{R}^{N \times d}$ are input and output features for the visual regions, $Conv_1(\cdot)$ is the CNN layer with kernel size 1, $DilatedConv_3(\cdot)$ is the dilated layer with kernel size 3. Each convolution layer is followed by the batch normalization with a ReLU activation. Empirically, stacking multiple modules typically capture more concise relations between regions. We thus stack $k$ modules to obtain $k + 1$ number of representations of the visual regions in different subspaces, denoted as $\{\mathbf{V}^{(0)}, \mathbf{V}^{(1)}, ..., \mathbf{V}^{(k)}\}$, where $\mathbf{V}^{(0)}$ is the original visual features ($\mathbf{V}$) and $\mathbf{V}^{(l)}$ is the output of layer $l$. We assume that each layer can learn the specific visual meaning, which will be used later to align with the corresponding textual answers.

We adopt the bidirectional GRU network to encode the input answer $A = \{a_1, a_2, ..., a_m\}$ with initial embedding $\{\mathbf{a}_1, \mathbf{a}_2, ..., \mathbf{a}_m\}$. We then concatenate the last hidden state of each directions to present the answer $A$ as a single vector $\mathbf{a} \in \mathbb{R}^d$. Similarly, to better align with visual regions, we also utilize multiple linear projections to project answer representation $\mathbf{a}$ into $k + 1$ number latent subspaces, denoted as $\{\mathbf{a}^{(0)}, \mathbf{a}^{(1)}, ..., \mathbf{a}^{(k)}\}$. Thus, for each subspace $l \in [0, k]$, we employ attention mechanism to learn the relevance between each visual region $\mathbf{v}_i^{(l)}$ ($\mathbf{v}_i^{(l)}$ is the $i$-th row of $\mathbf{V}^{(l)}$) and the answer $\mathbf{a}^{(l)}$ as follows:

$$s_i^{(l)} = \omega^\top \sigma(\mathbf{W}_i \mathbf{v}_i^{(l)} + \mathbf{a}^{(l)} + \mathbf{b}), \tag{3}$$

where $\mathbf{W}_i \in \mathbb{R}^{d \times d}, \omega \in \mathbb{R}^d, \mathbf{b} \in \mathbb{R}^d$ are the learnable weights, $\sigma(\cdot)$ is the Tanh$(\cdot)$ activation function, and $s_i^{(l)}$ is the similarity score capturing the semantic relevance between the region $r_i$ and the answer $A$. Then we take the average of relevance score in each layer and calculate the probability of being a visual hint:

$$p_{vh,i} = \sigma\left(\frac{\sum_{l=0}^{k} \mathbf{s}_i^{(l)}}{k+1}\right), \tag{4}$$

where $\sigma(\cdot)$ is the Sigmoid$(\cdot)$ function. Given the target visual hints $\mathcal{V}$, we apply focal loss [32] to optimize this module, which is denoted as $\mathcal{L}_{vh}$ as follows:

$$\mathcal{L}_{vh} = -\eta \sum_{r_i \in \mathcal{V}} p_{vh,i}^\lambda \log(p_{vh,i}) - \eta \sum_{r_i \notin \mathcal{V}} (1 - p_{vh,i})^\lambda \log(1 - p_{vh,i}), \tag{5}$$

where $\eta, \lambda$ are hyper-parameters.

### 2.2.2 Generating Question with Double Hints

After obtaining the visual hints, the next step is to generate the high-quality question under the guidance of double hints. We employ the attention-based hierarchical sequence decoder from [34] for the double-hints-based question generation step. This module consists of two LSTMs: 1) vision LSTM and 2) language LSTM. The vision LSTM is used to encode the global visual features, the input word embedding, and the answer embedding into hidden state $\mathbf{h}_1^t \in \mathbb{R}^d$, and the language LSTM is used to generate question text sequence. Note that the starting states are initialized by answer embedding. Between the two LSTMs, the attention module is employed guided by the predicted visual hints and the answer hints. Specifically, we first apply a simple pruning strategy to select the candidate visual regions and then identify the predicted visual hints to obtain $\mathbf{V}_{vh} \in \mathbb{R}^{T \times d}$ using the predictive model mentioned in the previous section. Next, we employ attention on the predicted visual hints $\mathbf{V}_{vh}$ with $\mathbf{h}_1^t$ (textural answer hint) to obtain the importance of the salient visual clues for generating the current word token in question. We refer the reader to Appendix C for more details.

We train it with the cross-entropy loss function denoted as $\mathcal{L}_{lm}$. As a result, the supervised loss function for the generator $G$ consists of visual hints prediction loss $\mathcal{L}_{vh}$ and language generation loss $\mathcal{L}_{lm}$ shown as follows:

$$\mathcal{L}_{sup} = \mathcal{L}_{lm} + \alpha \mathcal{L}_{vh}, \tag{6}$$

where $\alpha$ is the coefficient hyper-parameter.

## 2.3 D-step: Question-Answer Discriminator

Although the salient visual hints are predicted by our Generator, there is no clear way how to evaluate the quality of selected salient visual hints. In this section, we will introduce the design protocol for an effective discriminator in D-step by leveraging the signals from the generated questions.

In particular, the proposed discriminator is a binary classifier that takes the sample triplet $x = \{I, Q, A\}$, where $Q$ could be either the generated question or the ground-truth question, and then outputs a class label indicating whether the input triplet is from the ground-truth or the proposed generator. Therefore, the discriminator rewards the generator for identifying the most meaningful region of interest in order to leverage these visual hints to generate the best quality of questions.

Specifically, we employ bidirectional GRU to encode both question $Q$ and answer $A$ to its vector representations $\mathbf{q} \in \mathbb{R}^d$ and $\mathbf{a} \in \mathbb{R}^d$, respectively. Then we attend on the visual region features $\mathbf{V} \in \mathbb{R}^{N \times d}$ using question vector embeddings $\mathbf{q} \in \mathbb{R}^d$ to obtain the question-aware visual vector embeddings $\mathbf{v}_{attn}$. Next, we concatenate $\mathbf{v}_{attn}$ and answer vector $\mathbf{a}$ to generate the probability of being human-annotated triples as follows:

$$p = \sigma(F(\mathbf{v}_{attn} \parallel \mathbf{a})), \tag{7}$$

where $F(\cdot) : \mathbb{R}^{2d} \to \mathbb{R}$ is the linear projection, $\sigma(\cdot)$ is the sigmoid function and $\parallel$ is the simple concatenation. We then apply binary cross-entropy function to calculate the loss denoted as $\mathcal{L}_D$.

## 2.4 Training DH-GAN via Policy Gradient

Our model is not fully differentiable due to the procedure of choosing the next token based on the probability distribution, which makes it infeasible to propagate the gradients from the discriminator to the generator. Inspired by the sequence GAN [24, 53, 8], we adopt the policy gradient mechanism to handle this issue. Concretely, the reward score of the generated sample $x^-$, which is formulated by the discriminator, is used as a reward signal for the generator. Formally, given a sample $x$, we define the reward function as $R(x)$. We adopt the reinforcement learning (RL) algorithm [47] to maximize the expected reward of generated samples $x^-$:

$$J(\theta) = \mathbb{E}_{Q^- \sim P(Q^-|I,A)}(R(x^-)|\theta), \tag{8}$$

where $Q^-$ is the question generated by the generator and $\theta$ is the model parameter. The equation can be optimized by the likelihood ratio trick [47, 10, 24, 8]:

$$\nabla J \approx [R(x^-) - B(x^-)] \nabla \log P(Q^-|I, A) \tag{9}$$

where $\log P(Q^-|I, A)$ is the log-likelihood of the question generator, $R(x^-)$ is the reward of being "real" given a generated sample $x^-$, and $B(x^-)$ is the baseline which can reduce the variance of RL system while keeping unbiased. Obviously, how to select an appropriate baseline is one of the keys to learn a useful reward.

Inspired by the self-critical sequence training (SCST) [5, 42], we utilize the output of the test-time inference to normalize the rewards the generator meets. At each iteration, the generator will output two results: 1) the baseline output $\hat{Q}$ which is obtained by greedy search, that is, by maximizing the output probability distribution at each decoding step, 2) the sampled output $\overline{Q}$, generated by multinomial sampling, that is, each word token $\overline{q}_t \in \overline{Q}$ is sampled according to the likelihood $P(\overline{q}_t | I, A, \overline{q}_{<t})$ predicted by the generator. Following that, the Eq. 9 can be written as:

$$\nabla J \approx [R(\{I, \overline{Q}, A\}) - R(\{I, \hat{Q}, A\})] \nabla \log P(\overline{Q}|I, A) \tag{10}$$

Equally important, the exploration of the visual hints could lead to a better question generation. Therefore, for an individual region $r_i$, we determine whether it could be a visual hint according to the probability of $p_{vh,i}$ in Eq. 4. We denote the sampled visual hints as $\overline{\mathcal{V}}$. It is worth noting that to make the training stable, we apply the temperature $\tau$ on the logits before sigmoid in Eq. 4. Intuitively, a good exploration of the visual hints would be beneficial to improve the quality of the generated questions. Thus we apply the same reward gain in Eq. 10, which can be written as:

$$\begin{aligned} \nabla J &\approx [R(\{I, \overline{Q}, A\}) - R(\{I, \hat{Q}, A\})] \nabla \log P(\overline{Q}|I, A) \\ &= [R(\{I, \overline{Q}, A\}) - R(\{I, \hat{Q}, A\})] \nabla [\log P(\overline{Q}|I, A, \overline{\mathcal{V}}) + \log P(\overline{\mathcal{V}}|I, A)]. \end{aligned} \tag{11}$$

### 2.4.1 Reward Function

One of the key factors of Reinforcement Learning is to design a proper reward function. In order to generate syntactically and semantically meaningful questions, we first consider adopting one of the automatic evaluation metrics (i.e. BLEU@4 [37]) for measuring the syntactic quality of the generated questions as part of the optimization target. Since it is well known that the BLEU score cannot faithfully reflect the semantic quality, we further select the discriminator's probability output as a reward to measure the semantic quality. Note that, the larger $D(x)$ is, the more "real" question $Q$ is. Therefore, we define the hybrid reward function as follows:

$$R(x) = BLEU(Q) + \epsilon D(x), \tag{12}$$

where BLEU($\cdot$) is the *generation reward*, and $D(\cdot)$ is the *semantic reward* given the ground-truth question, respectively. Moreover, $\epsilon$ is the coefficient balancing these two rewards.

### 2.4.2 The Loss Functions of DH-GAN for VQG

**Generator Loss.** The loss function of the generator derived from Eq. 11 can be written as:

$$\mathcal{L}_{rl} = -[R(\{I, \overline{Q}, A\}) - R(\{I, \hat{Q}, A\})][\log P(\overline{Q}|I, A, \overline{\mathcal{V}}) + \beta \log P(\overline{\mathcal{V}}|I, A)], \tag{13}$$

where $\beta$ is the hyper-parameter, $\log P(\overline{Q}|I, A, \overline{\mathcal{V}})$ is the question generation loss with target question $\overline{Q}$ in Sec. 2.2.2 and $\log P(\overline{\mathcal{V}}|I, A)$ is the visual hints prediction loss given target visual hints $\overline{\mathcal{V}}$ in Sec. 2.2.1. Practically, we find that it is unstable to update the generator by minimizing the loss $\mathcal{L}_{rl}$. Thus we combine both the teacher-forcing loss $\mathcal{L}_{sup}$ in Eq. 6 and the reinforcement loss as:

$$\mathcal{L}_G = \gamma \mathcal{L}_{rl} + (1 - \gamma)\mathcal{L}_{sup}, \tag{14}$$

where $\gamma$ is a scaling factor controlling the trade-off between teacher-forcing loss and RL loss.

**Discriminator Loss** After updating the generator for some steps, we will re-train the discriminator to classify the generated and the original samples better as follows:

$$\mathcal{L}_D = -\log D(x^+) - \log(1 - D(x^-)) \tag{15}$$

## 3 Experiments

In this section, we conduct an extensive set of experiments to demonstrate the effectiveness of the proposed method compared with existing state-of-the-art methods[1].

**Dataset.** We conduct the experiments on the VQA2.0 [3] and COCO-QA [39] datasets. Following [20], the visual hints are generated by rule-based similarity matching (see Appendix E.1.1 for more details). Clearly, the original visual hints are too noisy to guide the question generation. To alleviate this issue, we introduce a simple pruning mechanism to decrease the number of candidates while preserving the key information. After pruning, we will keep at most $m$ visual regions to be visual hints (see Appendix E.1.2). After pre-processing, the VQA2.0 has 278707/135584, and the COCO-QA dataset has 58979/29017 examples for training/validation split, respectively.

**Pre-processing.** For each input image, we employ pre-trained Faster-RCNN [40] for visual region detection and feature extraction (fc6) (see Appendix E.1.3). For text data, we truncate the questions longer than 20 words and build the vocabulary on the words with at least 3 occurrences. Since the test splits (for these two datasets) are not open for the public, we divide the validation set to 10% validation split and 90% test split.

**Baseline Methods.** We compare against the following baselines in our experiments: i) I2Q* [49]: a standard CNN-RNN model in image caption that maps the image to the questions directly. ii) IT2Q [22]: a CNN-RNN based model which uses answer type as side information. iii) IMVQG* [22]: a variational method that maximizes the mutual information between question, image, and the expected answer. iv) Dual* [28]: an advanced model that combines the VQA and VQG tasks by

---

[1]The code and data for our model are provided for research purposes: DH-GAN for VQG Github Repo.

Table 1: Results on VQA2.0 and COCO-QA val set. All metrics are in % and B@4-BLEU@4, C-CIDEr, M-METEOR, R-ROUGE, S-SPICE.

| Dataset | VQA 2.0 | | | | | COCO-QA | | | | |
|---------|------|--------|-------|-------|-------|-------|--------|-------|-------|-------|
| Method | B@4 | C | M | R | S | B@4 | C | M | R | S |
| I2Q | 9.02 | 63.21 | 13.89 | 35.33 | 18.04 | 14.71 | 107.90 | 13.71 | 38.32 | 18.65 |
| IT2Q | 18.41 | 134.88 | 19.90 | 45.71 | 22.90 | 18.04 | 135.23 | 17.34 | 46.76 | 22.21 |
| IMVQG | 19.72 | 149.28 | 20.43 | 47.20 | 23.10 | 21.16 | 156.76 | 18.93 | 46.89 | 24.21 |
| Dual | 19.90 | 151.60 | 20.60 | 47.00 | 23.21 | 21.48 | 153.32 | 18.93 | 47.03 | 24.34 |
| Radial | 21.87 | 162.92 | 22.22 | 48.65 | 25.34 | 22.63 | 168.29 | 19.73 | 47.71 | 26.71 |
| DH-VQG | 22.43 | 180.55 | 22.57 | 49.36 | 27.40 | 23.15 | 175.18 | 20.04 | 47.84 | 27.63 |
| **DH-GAN** | **23.71** | **191.06** | **22.91** | **50.53** | **28.18** | **23.52** | **186.65** | **20.44** | **48.61** | **28.32** |

dual learning. v) Radial* [50]: a strong baseline that evolves an answer-centric radial graph with GNN encoding to ask questions. vi) DH-VQG [20]: a recently proposed baseline that uses very noisy double-hints as side information to generate referential questions. The detailed descriptions of these baselines are provided in Appendix D. Experiments on baselines followed by * are conducted using the codes originally released by authors.

**Evaluation Metrics.** Following previous works [22, 50, 20], we adopt the standard linguistic measures including BLEU [37], CIDEr [45], METEOR [4], ROUGE-L [31] and SPICE [1]. For visual hints prediction, we adopt F1 metric. These scores are calculated by officially released evaluation scripts[1].

## 3.1 Results Analysis and Human Evaluation

Table 1 shows the results comparing our proposed method against other state-of-art baselines on various automatic evaluation metrics. Note that, due to limited space, we omit the reports of the result performance of the visual-hints only (please see these details in Appendix F). There are two observations worth noting here. First, our proposed DH-GAN method for VQG consistently outperforms other state-of-the-art methods by a large margin, highlighting the importance of double visual hints and textual answer hints. Second, compared with recently proposed DH-VQG, DH-GAN improves over DH-VQG by a noticeable margin thanks to the proposed GAN method that could effectively identify the salient visual regions associated with the answer hints.

Furthermore, we conduct a human evaluation study to assess the quality of the questions generated by our proposed DH-GAN method, the ground truth (GT), and the baseline-Radial on three different dimensions in terms of syntax, semantics, and relevance metrics, respectively. In addition, to investigate the effectiveness of DH-GAN,

Table 2: Human-study Results on VQA2.0.

| Method | Syntax | Semantics | Relevance |
|--------|--------|-----------|-----------|
| Radial | 4.43 (0.48) | 4.44 (0.6) | 3.48 (0.59) |
| Generator | 4.67 (0.37) | 4.66 (0.48) | 3.86 (0.6) |
| DH-GAN | 4.68 (0.32) | 4.65 (0.42) | 4 (0.58) |
| GT | **4.7 (0.33)** | **4.67 (0.44)** | **4.31 (0.53)** |

we also compare the results with only pre-trained generator without the proposed discriminator (abbr: Generator). The results are shown in Table 2. It is clear to see that our model outperforms all strong baseline methods on almost all three metrics, especially on relevance score (the closest one to the ground-truth), which highlights how important to identify the most important visual region of interest in order to reduce the noises. Our human evaluation details are discussed in Appendix H.

## 3.2 Ablation Study

We further conduct the ablation study to evaluate the importance of each key component of the proposed DH-GAN method on the VQA 2.0, including visual hints, visual hints sampling, textual answer hints and answer type. Note that, in order to make the comparison consistent and clear, we use the Generator model (from our DH-GAN) as baseline (abbr: Generator) and investigate the performance effect of each component by turning off or adding modules. The ablation study results on the VQA2.0 validation set are shown in Table 3.

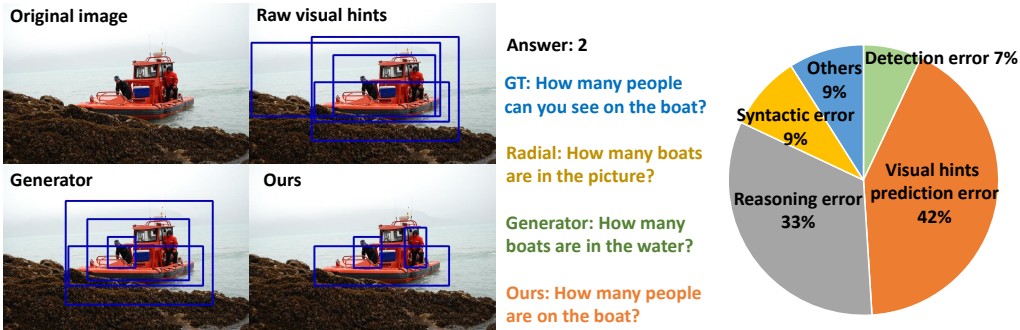

Figure 2: Case study (left) and Error analysis (right)

There are several important observations worth highlighting here. First, by turning off the visual hints (- visual hints), the performance of model drops nearly 3.2% (BLEU@4). Clearly, the visual hints are indeed helpful for generating high-quality questions. When we further replace the answer hints with only answer type, the performance continues to drop rapidly by 17.5% (BLEU@4), which indicates that the answer hints carry much more helpful information than the answer type.

Second, comparing our DH-GAN (full model) with the baseline Generator, we can see that the performance improves significantly by 4.9% (BLEU@4), which demonstrates the effectiveness of the GAN framework for generating informative and referential questions by identifying the most important salient visual regions. Furthermore, we notice that by removing the visual hints sampling module (+GAN - visual hints sampling), the performance drops from 23.71 to 23.53 (BLEU@4), which indicates that the visual hints sampling strategy is beneficial to the VQG task by encouraging the model to explore more visual hints.

Table 3: Ablation Results on VQA2.0. Note that "+" denotes having this component and "-" means removing this component.

| Method | B@4 | M | R |
|---|---|---|---|
| DH-GAN (full model) | 23.71 | 22.91 | 50.53 |
| + GAN
- visual hints sampling | 23.53 | 22.87 | 50.45 |
| Generator | 22.60 | 22.69 | 49.77 |
| - visual hints | 21.87 | 22.05 | 49.14 |
| - visual/answer hints
+ answer type | 18.63 | 20.01 | 45.43 |

### 3.3 Case Analysis and Error Analysis

Next, we perform the case study to illustrate the superiority of our method compared with other baselines: 1) Radial, 2) generator without DH-GAN (abbr: Generator). Specifically, we visualize the visual hints: 1) generated with pre-processing visual hints (abbr: Raw visual hints), 2) predicted with only Generator (abbr: Generator), and 3) predicted with full DH-GAN (abbr: Ours). As shown in Fig. 2 (left), we can find that the raw visual hints obtained in pre-processing are quite noisy. Such noisy visual hints fail to provide the clear supervised signal to guide the generator, leading to the imprecise prediction of the generator. In contrast, with the help of DH-GAN, the generator can predict more accurate and referential ones, which faithfully guides the generator to predict precise and high-quality questions. Please refer to Appendix I for more detailed case studies.

Moreover, we further investigate the failures of our model by pinpointing various bad cases. The results are shown in Fig. 2 (right). Since it is very time-consuming to analyze the errors of every example, we qualitatively categorize them into five different error groups. Clearly, even with our DH-GAN method, most of the bad cases are caused by the first error type - visual hint prediction error, which is not surprising since we have no human annotations on visual hints. The second-largest error group is about reasoning error (i.e., yes/no), which indicates the considerable improvement potential by truly understanding the logical polarity in the answer. The remaining errors are mainly from detection error and syntactic error, which can be improved with better object detection tools and language models. More detailed examples and analyses are provided in Appendix J.

### 3.4 Data Augmentation and Zero-shot VQA

To investigate the effect of VQG for real world applications, we apply the VQG to the visual question answering (VQA) under two settings: 1) as a data augmentation method, 2) as a solution for zero-shot VQA.

Data augmentation is one of the most important applications of VQG. In this section, we use our proposed DH-VQG method to generate more questions for training a better VQA model. We employ two existing VQA models: 1) Bottom-Up Top-Down (abbr: BUTD [2]) and 2) the Bilinear Attention Network (abbr: BAN [21]). We compare our DH-VQG with Radial (a strong baseline) to demonstrate the effectiveness of the QG-driven data augmentation.

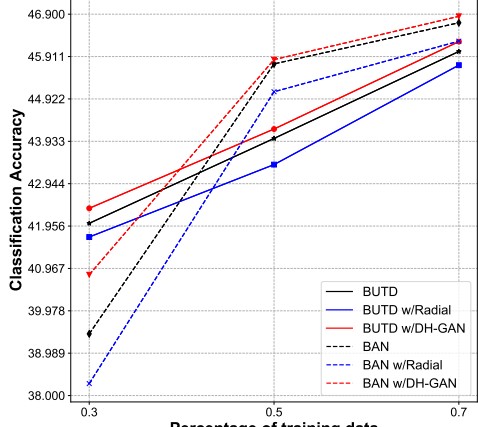

Figure 3: Data augmentation

Specifically, we split the VQA2.0 train-set (the same as the VQG dataset) to $x \in \{0.3, 0.5, 0.7\}$. The VQA models are trained only on the $x$ part, while the other VQA variants are trained on the combination of the golden $x$ part and the questions generated by VQG models. The results are shown in Fig. 3. Surprisingly, we observe that the Radial augmented VQA models perform inferior to the single VQA models. We think that the generated questions contain noises, which may mislead the VQA model in some ways [50]. Notably, our DH-GAN model outperforms all the baselines consistently, demonstrating the high-quality of the generated questions.

In addition, we introduce the VQG as an effective solution for the zero-shot VQA (ZS-VQA) problem. Following the practice of [50, 29], we first obtain the *zero-shot words* and then construct the *zero-shot dataset* for VQA 2.0. Concretely, we first find the same words between the questions and answers in the original VQA dataset (including the training and testing) by filtering out stop words. Then we uniformly sample 10% words from them and obtain the zero-shot words. Secondly, we pick up all samples whose answers contain zero-shot words

Table 4: VQG for Zero-shot VQA Results on VQA2.0.

| Method | BUTD | BAN |
|---|---|---|
| w/o. VQG. | 0% | 0% |
| Radial VQG | 18.15% | 18.37% |
| DH-GAN | 19.21% | 19.93% |

and construct the ZS-VQA testing set. Finally, the ZS-VQA training set which has no overlap with the ZS-VQA testing set can be constructed. Similar to the above data augmentation experiment, we employ two VQA methods: 1) BUTD, 2) BAN, and two VQG methods: 1) Radial, 2) DH-GAN. Technically, for each variant, we use the VQG method to generate the questions and produce additional training data, which can be used to train the VQA model. Following [50], since ZS words appear in the question, they can be found in the images and thus we can treat them as the target answers to generate questions to help the VQA model to predict the zero-shot answer. As shown in Table 4, we observe that both BUTD and BAN achieve 0% in the ZS-VQA test set. This is because without the help of VQG, both VQA models cannot see the ZS-words during the inference. However, when we leverage various VQG methods, both the VQA models achieve remarkable improvements, which demonstrate that the VQG can be very beneficial in solving the ZS-VQA problem. Furthermore, we observe that the DH-GAN variants outperform the Radial variants, which proves the superiority of our proposed method in generating high-quality QA pairs.

## 4   Related Work

**Visual Question Generation.**   Visual question generation [35, 7, 38, 25, 48] is a promising and attractive task in the vision and language domain. Jain et al. [18] firstly introduced the variational autoencoder (VAE) based framework to generate diverse questions. Krishna et al. [22] introduced the answer-type as side information based on the variational model to generate more goal-driven results. Furthermore, Li et al. [28], Shah et al. [43] combined the visual question answering (VQA) and visual question generation tasks by dual learning mechanism. Liu et al. [33] further formulated the VQG problem as the inverse of the VQA task and regard the type-specific partial question as side information to guide the training process. Recently, Xu et al. [50] adopted the GNN method to generate answer-specific questions. Kai et al. [20] introduced the double-hints, leveraging both textual answers and visual regions to faithfully guided the generating process, which made a valuable step to address the one-to-many mapping issue.

**Generative Adversarial Network for Text Generation.** The generative adversarial network has achieved great success in computer vision [12, 51, 57, 13, 27, 54, 6, 9]. However, compared with images, the language is discrete, making it infeasible for gradients propagating from discriminator to generator. Li et al. [24], Yu et al. [53] firstly proposed the sequence GAN, which employs the Reinforcement Learning mechanism to provide feedback obtained from the discriminator as a reward to the generator. Fedus et al. [8] proposed the MaskGAN, which trains the model on a text fill-in-the-blank or in-filling task to reduce the training instability and model drop problem. Despite the RL-based methods [26, 5, 11], Kusner and Hernández-Lobato [23] attempted to find a continuous approximation of discrete sampling by the Gumbel Softmax mechanism. In addition, Zhang et al. [56] proposed the continuous kernelized discrepancy function to approximate the non-differentiable sampling function in the original GAN.

**Noisy Supervision.** Learning from noisy data with noisy supervision has attracted significant amount of interests in recent years [30, 15, 14, 41, 16, 55, 52]. Li et al. [30] proposed a unified framework to distill the knowledge learned from clean dataset to learn a better model from noisy labels. Han et al. [15] proposed the "co-teaching" learning diagram for combating with noisy labels. They employ two network and let them teacher each other simultaneously. Han et al. [14] proposed a human-assisted approach called "Masking" which conveys the external rules to speculate the noise transition matrix effectively. Ren et al. [41] further studied the text classification task with rule-induced weak labels. They designed a label denoiser to estimate the the source reliability to learn from noisy data. Different from these methods which mostly focus on classification task with noisy target labels, we focus on a generation task under noisy supervision.

## 5 Conclusion

In this paper, we present a novel learning approach for double-hints based VQG, which can be cast as a weakly supervised learning problem with noises. In particular, we propose a Double-Hints guided Generative Adversarial Networks (DH-GAN), which consists of a double-hints based question generator and a question-answer-aware discriminator. Thanks to DH-GAN, the generator and discriminator undergo an adversarial process to encourage the generator to identify the salient visual regions of interest so that the high-quality question can be generated, which are indistinguishable from the ground truth questions by the discriminator. Our extensive experiments demonstrate the effectiveness of our proposed model.

## 6 Acknowledge

This work has been supported in part by the National Key Research and Development Program of China (2018AAA0101900), Zhejiang NSF (LR21F020004), Chinese Knowledge Center of Engineering Science and Technology (CKCEST).

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
