# A   Appendix

Optionally include extra information (complete proofs, additional experiments and plots) in the appendix. This section will often be part of the supplemental material.

# B   DH-GAN Algorithm

In this section, we show the detailed training procedure in summary. As shown in Algorithm 1, before training adversarially, we pre-train the generator to ensure it can produce reasonable questions. Then we regard the samples generated by the generator with sampling as negative samples and pre-train the discriminator.

During the DH-GAN's training process, the generator and discriminator are trained iteratively. The generator is encouraged to generate deceptive samples to fool the discriminator, while the discriminator needs to keep pace with the generator.

---

**Algorithm 1** DH-GAN

**Input:** Generator $G$; Discriminator $D$; VQG dataset $\mathcal{S}$
1: Pre-train generator $G$ by $\mathcal{L}_{sup}$
2: for pretrain-discriminator-steps do
3:     Sample $\mathcal{X}^+ = \{x_1^+, x_2^+, ..., x_n^+\}$ from dataset $\mathcal{S}$
4:     Generate samples $\mathcal{X}^- = \{x_1^-, x_2^-, ..., x_n^-\}$ from generator $G$
5:     Updating discriminator $D$ by $\mathcal{L}_D$
6: **repeat**
7:     for generator-steps do
8:         Sample $\mathcal{X}^+ = \{x_1^+, x_2^+, ..., x_n^+\}$ from dataset $\mathcal{S}$
9:         Generate samples $\mathcal{X}^- = \{x_1^-, x_2^-, ..., x_n^-\}$ from generator $G$
10:         Updating generator $G$ by $\mathcal{L}_G$
11:     for discriminator-steps do
12:         Sample $\mathcal{X}^+ = \{x_1^+, x_2^+, ..., x_n^+\}$ from dataset $\mathcal{S}$
13:         Generate samples $\mathcal{X}^- = \{x_1^-, x_2^-, ..., x_n^-\}$ from generator $G$
14:         Updating discriminator $D$ by $\mathcal{L}_D$
15: **until** DH-GAN converges

---

# C   The Details of Double-hints-guided Question Decoder

The double-hints-guided question decoder consists of two LSTM: 1) vision LSTM and 2) language LSTM and a double-hints-guided attention module between them.

**Vision LSTM**   Technically, at time step $t$, we first adopt the vision LSTM to encode the global visual feature and the input word embedding $\mathbf{x}_t$ into the hidden state $\mathbf{h}_1^t \in \mathbb{R}^d$. $d$ is the decoder's hidden size.

$$\mathbf{h}_1^t = \text{LSTM}(\mathbf{v}_{pool} \parallel \mathbf{x}_t, \mathbf{h}_1^{t-1}), \tag{1}$$

where $\mathbf{v}_{pool} \in \mathbb{R}^d$ is the mean-pooling of the image region features $\mathbf{V}$, $\mathbf{x}_t$ is the input word's embedding vector, $\cdot \parallel \cdot$ is the concatenation operation, and $\mathbf{h}_1^{t-1}$ is the previous step's hidden state.

**Double-hints-guided Attention**   The double-hints-guided attention module then dynamically attends on the visual region features with the guidance of double hints. Technically, we first prune the visual regions by the predicted visual hints and then apply attention by $\mathbf{h}_1^t$ (textural hint). Therefore, we define it as follows:

$$\begin{aligned} \mathbf{V}_{vh} &= \text{VisualHintMask}(\mathbf{V}) \\ \mathbf{h}_r &= \text{Attention}(\mathbf{V}_{vh}, \mathbf{h}_1^t) \end{aligned} \tag{2}$$

where *VisualHintMask* is to mask off the non-visual-hint objects, $\mathbf{V}$ is visual regions' embedding and *Attention* is the classic attention mechanism [1]. It is worth noting that there is one special case that no region is predicted as visual hints. We will reveal every region under this condition.

**Language LSTM**   The language LSTM will encode the vision LSTM's and double-hints-guided attention's results to generate the words.

$$\mathbf{h}_2^t = \text{LSTM}(\mathbf{h}_r \parallel \mathbf{h}_1^t, \mathbf{h}_2^{t-1}) \tag{3}$$

where $\cdot \parallel \cdot$ is the concatenation operation and $\mathbf{h}_2^{t-1}$ is the previous time step's hidden state. We project the $\mathbf{h}_2^t$ to the vocabulary space with softmax operation to generate the word.

# D   The details of the baseline models

**I2Q**   It means generating the questions without any hints. We adopt the classic image caption *show attend and tell* method [12].

**IT2Q**   It means generating questions with answer types. Technically, we adapt the image caption model *show attend and tell* [12], which takes the input from the joint embedding of image and answer type to predict questions with answer-type side information. Since there are no additional answer-type annotations in the original datasets, we follow [6, 3] and annotate them by hand. We will discuss in baseline *IMVQG*.

**IMVQG [6]**   This is a variational baseline that maximizes the mutual information among the generated questions, the input images, and the expected answers. Note that they annotate 80% training samples' answer-type and drop the remain. To fit our training data, for the VQA2.0 dataset, we annotate training samples' answer-type, which are missing in their training data, as "other". As for COCO-QA, we follow [3] and annotate the answer-type by hand since there are only 430 answers.

**Dual [7]**   This is a competitive baseline that employs dual learning to train the VQG task together with VQA task. Specifically, they formulate the VQG task as a dual task of VQA task based on MUTAN architecture and train them by cycle consistency to enhance both the VQG and VQA's performance.

**Radial [13]**   This is a strong baseline for the VQG task, which adopts answers as side information. Technically, they build the answer-radial object graph and employ GNN based method to learn the embedding. Then they adopt the graph2seq method to generate the questions.

**DH-VQG [3]**   This is the latest baseline for VQG with double hints. They propose the rule-based similarity method to obtain the visual hints. Technically, they first align the visual regions with double hints and then adopt the graph2seq framework to generate the questions.

# E   Implementation Details

## E.1   Dataset and Pre-processing

### E.1.1   Annotating Visual Hints

Following [3], we adopt the rule-based similarity matching technique to obtain visual hints of the original training samples (a sample refers to an image, a question, and an answer) automatically. Firstly, we adopt object detection tools [9] to generate $N$ visual regions. Each region $r_i \in \mathcal{R}$ is associated with class attribute and confidence score. Then we use Stanford CoreNLP [8] to find the noun-words in both questions and answers. The visual regions' class attributes and noun words are all initialized by GloVe embedding with mean-pooling. The visual region $r_i \in \mathcal{R}$ is regarded as a visual hint iff its' L2 distance with any noun-words is smaller than the threshold $\mu$. We denote the obtained visual hint candidates set as $V_{candidate}$. Examples of generated visual hints are shown in Fig. I with title: *Raw visual hints w/o. pruning*.

When annotating the visual hints by the proposed rule-based similarity matching technique, two special cases can lead to no matched objects: (1) there are exactly no visual hints (e.g., Q: Is this a cat? A: No) (2) the error caused by the detection model or the NLP tools leads to no matched visual hints [3]. Following [3], for the first case, we will keep them. For the second case, we will drop them

due to technical drawbacks. What's more, for an image-answer pair that have multiple questions, we will randomly reserve one [3].

Specifically, we find that there are some class attributes in Visual Genome that can't be represented by GloVe. Thus for each class phrase, we replace it with the closest term in GloVe. Such mapping is attached in Table 1.

Table 1: The mapping of class attributes in VG and glove respectively

| Class attribute in VG | Class attribute in GloVe |
| --- | --- |
| ceiling fan | fan |
| birthday cake | cake |
| skateboard ramp | ramp |
| towel rack | rack |
| tree branch | branch |
| tile floor | floor |
| ski jacket | anorak |
| tennis court | court |
| rock wall | wall |
| tennis racket,tennis racquet | racquet |
| toilet brush | brush |
| wii remote | remote-control |
| brocolli | broccoli |
| sandwhich | sandwich |
| skiis | skis |
| kneepad | kneecap |

### E.1.2 The Details of Visual Hints Pruning

Formally, we assume that the region with familiar class (i.e., the class attribute shared by lots of regions) but low confidence score is less important and should be pruned. Following this assumption, we select at most $m$ ($m < |V_{candidate}|$ for most cases) regions as the pruned visual hints (denoted as $V$) according to the class attributes and the confidence scores as follows:

(1) In the beginning, we will choose at most $m$ the class attributes from the candidates $V_{candidate}$. First, we sort the candidates by the confidence score in descending order. Second, we scan the candidates $V_{candidate}$ in order and record at most $m$ class attributes (no repetition). We denote the selected classes as $\mathcal{C}$ (w.r.t $|C| \leq m$).

(2) Then we will select at most $m$ regions according to the selected classes $\mathcal{C}$ and the confidence score attribute. For each selected class $c \in \mathcal{C}$, we will pick out one region with the largest confidence score (without replacement). We will repeat this procedure until $m$ regions are selected. Note that if $m >= |V_{candidate}|$ meets, all visual region candidates are the final visual hints.

### E.1.3 The Details of Pre-processing Images

We employ Faster-RCNN [9] with ResNeXt-101 backbone [11] implemented by Detectron2 [10], which is pre-trained on Visual Genome [5], to extract visual regions from images. Following the previous works[1], we extract 36 visual regions for all images with different NMS settings.

### E.2 The Setting of Model and Hyper-parameters

In the pre-processing, the threshold $\mu$ is set 5.7 according to [3]. For the visual hints pruning, we set the maximum number $m$ to 4. The word embeddings, whose dimension is set to 512, are initialized in random. The hidden size of GRU encoders is also set to 512. The hidden size of the double-hints-guided question decoder (both LSTM and attention module) is set to 1024. The unmentioned hidden sizes are all set to 1024.

---

[1]Please refer to the implements.

As for the visual hints predictor, we employ 3 layers of the reasoning module. The $\eta$ and $\lambda$ in focal loss (Eq. 5) are 4 and 2, respectively. The other important hyper-parameters are shown the Table 2 for both VQA2.0 and COCO-QA datasets.

During training, we adopt Adam optimizer [4] for the generator and AdaGrad [2] optimizer for the discriminator, respectively. During the pre-training stage, we set the learning rate to 0.0005 for the generator and 0.001 for the discriminator. During the DH-GAN's training process, we set the initial learning rate to 0.00001 for both the generator and discriminator. We conduct our experiments on 2 2080Ti GPUs on a single computer.

Table 2: The details of hyper-parameters for both VQA2.0 and COCO-QA datasets

| Dataset | $\gamma$ | $\tau$ | $\epsilon$ | $\alpha$ | $\beta$ |
|---------|------|------|------|------|-------|
| VQA2.0 | 0.99 | 0.3 | 0.5 | 0.01 | 0.001 |
| COCO-QA | 0.99 | 0.2 | 0.4 | 0.01 | 0.001 |

## F   The Details of Results

See Table 3 and Table 4 for full results.

Table 3: Results on VQA2.0 val set. All metrics are in %.

| Method | BLEU@4 | CIDEr | METEOR | ROUGE | SPICE | F1 |
|--------|--------|-------|--------|-------|-------|-----|
| I2Q | 9.02 | 63.21 | 13.89 | 35.33 | 18.04 | - |
| IT2Q | 18.41 | 134.88 | 19.90 | 45.71 | 22.90 | - |
| IMVQG | 19.72 | 149.28 | 20.43 | 47.20 | 23.10 | - |
| Dual | 19.90 | 151.60 | 20.60 | 47.00 | 23.21 | - |
| Radial | 21.87 | 162.92 | 22.22 | 48.65 | 25.34 | - |
| DH-VQG | 22.43 | 180.55 | 22.57 | 49.36 | 27.40 | 50.17 |
| Ours | **23.71** | **191.06** | **22.91** | **50.53** | **28.18** | 51.72 |

Table 4: Results on COCO-QA val set. All metrics are in %.

| Method | BLEU@4 | CIDEr | METEOR | ROUGE | SPICE | F1 |
|--------|--------|-------|--------|-------|-------|-----|
| I2Q | 14.71 | 107.90 | 13.71 | 38.32 | 18.65 | - |
| IT2Q | 18.04 | 135.23 | 17.34 | 46.76 | 22.21 | - |
| IMVQG | 21.16 | 156.76 | 18.93 | 46.89 | 24.21 | - |
| Dual | 21.48 | 153.32 | 18.93 | 47.03 | 24.34 | - |
| Radial | 22.63 | 168.29 | 19.73 | 47.71 | 26.71 | - |
| DH-VQG | 23.15 | 175.18 | 20.04 | 47.84 | 27.63 | 52.24 |
| Ours | **23.52** | **186.65** | **20.44** | **48.61** | **28.32** | 53.40 |

## G   More Experimental Results of Hyper-parameters

To further study the effect of hyper-parameters, we conduct comprehensive experiments with parameters varying in a certain range. The results are shown in Fig. 1 and Fig. 2.

- Firstly, we study the effect of $\gamma$ balancing the RL loss and teacher-forcing loss in Eq. 14. As shown in Fig. 1 (a), the model performs the best when $\gamma$ is 0.99. Specifically, we observe that when $\gamma$ is 1 (i.e., no teacher forcing loss), the performance drops rapidly, demonstrating that the combination of RL loss and teacher forcing loss is effective. And when $\gamma$ decreases from 0.99 to 0.9, we observe that the performance drops. Because when the teacher forcing rate is large, the exploration (i.e., the sampling in the RL) is suppressed, which can harm the system.

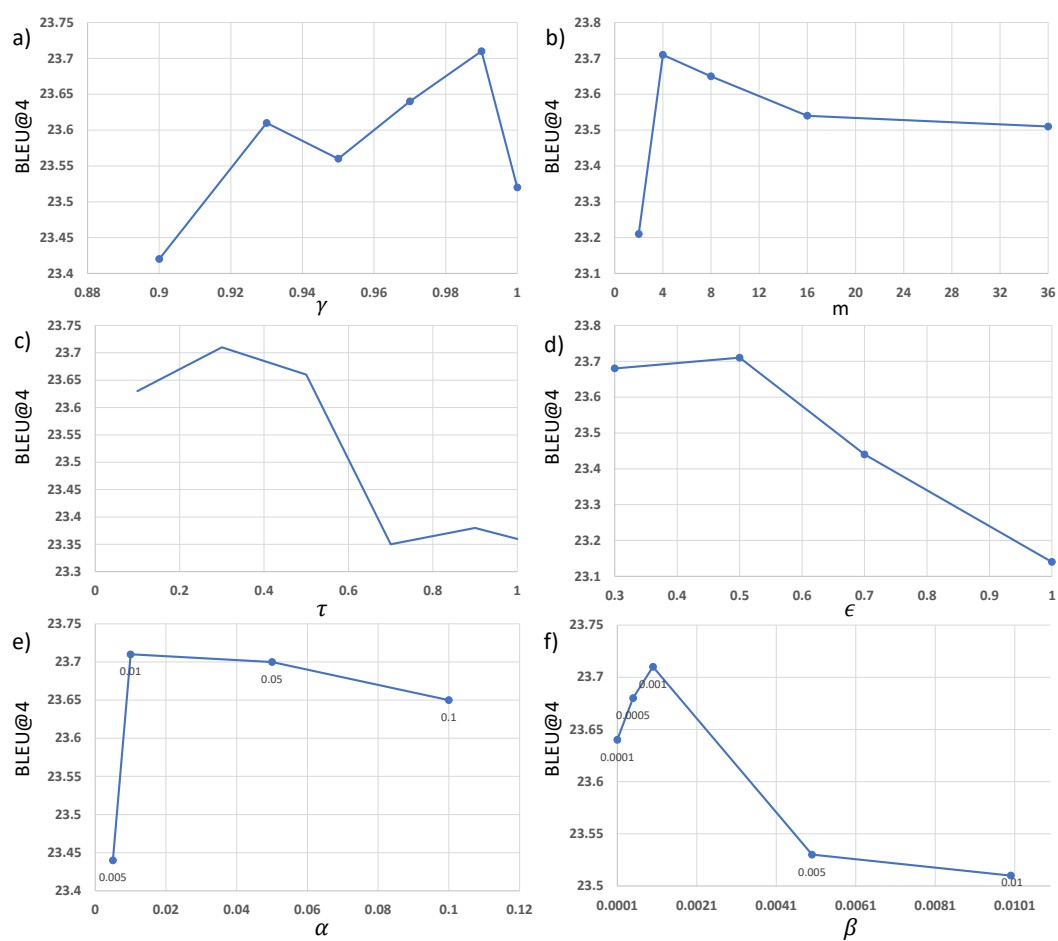

Figure 1: The analysis of different hyper-parameters.

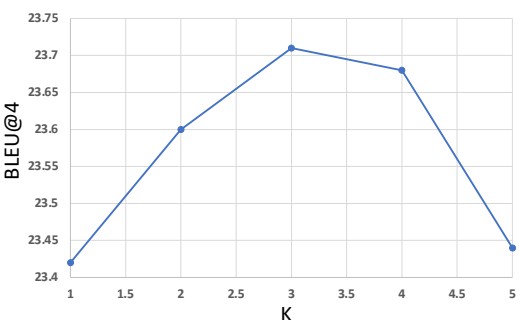

Figure 2: The analysis of parameter K.

- Secondly, we study the effectiveness of the visual hints pruning. As shown in Fig. 1 (b), by varying the maximum number of visual hints (i.e., $m$), we observe that the model performs the best when it is 4. When it is too small, the performance drops rapidly. Because many visual hints which are vital to the VQG may be pruned. When it is too large, the performance also drops because the visual hints are too noisy to guide the question generation procedure faithfully.

- Thirdly, we study the effect of temperature $\tau$ in visual hints sampling. As shown in Fig. 1 (c), we observe that the model performs the best when it is 0.3. If $\tau$ is too large, the

126 distribution of the probability is too soft, which leads to numerous explorations of visual
127 hints. It can hurt the performance. If the temperature is too small, the distribution becomes
128 too hard, which can suppress the exploration of visual hints.

129 • Fourthly, we study the effect of $\epsilon$ in the reward function balancing the *generation quality*
130 *reward* and the *semantic quality reward* in Eq. 12. As shown in Fig. 1 (d), The model
131 performs the best when $\epsilon$ is 0.5.

132 • Fifthly, we study the effect of $\alpha$ and $\beta$ balancing the visual hints prediction loss and language
133 generation loss in Eq. 6 and Eq. 13, respectively. As shown in Fig. 1 (e) and (f), we observe
134 that the model performs the best when $\alpha$ is 0.01 and $\beta$ is 0.001.

135 • Finally, we study the effect of $K$, which is the number of modules in visual hints generator.
136 As shown in Fig. 2, we observe that the model performs the best when $K$ is 3.

## H    The Details of Human Evaluation

138 In this section, we will discuss the detail of human evaluation on the VQA2.0. Following [**?** 3],
139 we conduct a small-scale human evaluation on the test split for four systems: 1) the ground truth
140 results (abbr: GT), 2) our DH-GAN results (abbr: DH-GAN), 3) the generator without GAN's
141 results (abbr: Generator), 4) the 'Radial' baseline's results (abbr: Radial). We randomly select 50
142 examples (each example contains the raw image, answer, and question) for each system and ask 5
143 human evaluators to give feedback on the quality of the randomly selected questions. In each example,
144 given a triple containing a raw image, a target answer, and an anonymized system's output, they are
145 asked to rate the quality by answering the following three questions: 1) is the question syntactically
146 correct? 2) is the question semantically correct? 3) is the question relevant to the image and the
147 answer pair? For each question, they are asked to rate from 1 to 5. The standard is: 1. Not acceptable,
148 2. Marginal, 3. Acceptable, 4. Good, 5. Excellent. In practice, we develop software to feed the
149 examples and collect the evaluation results automatically. The screenshot is attached in Fig. 3.

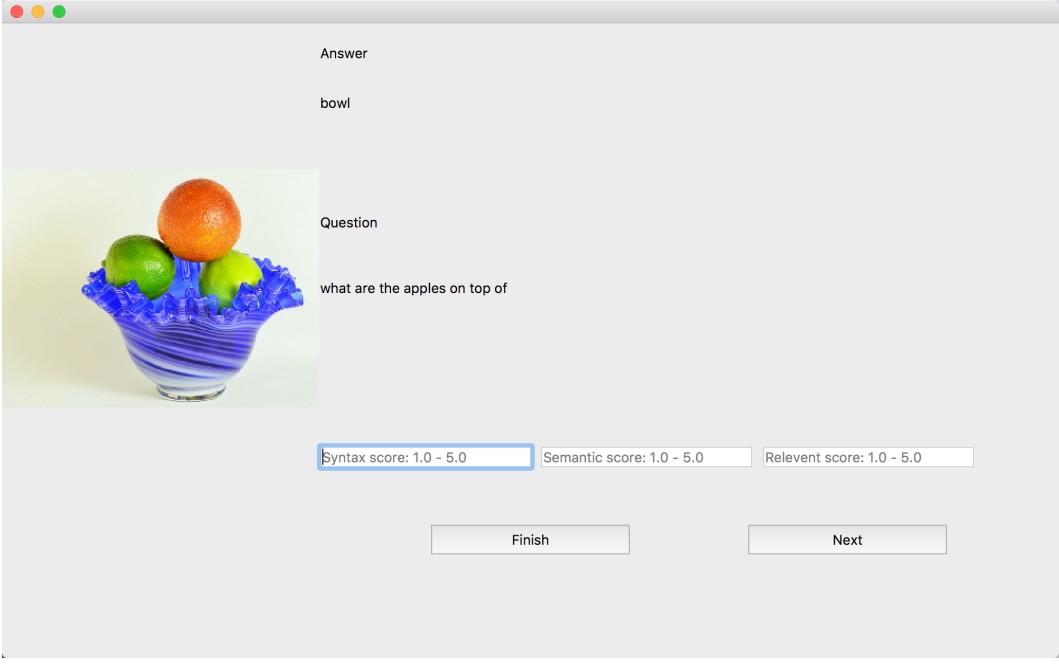

Figure 3: The screenshot of human evaluation software.

# I   More Examples of Case Study

In this section, we present more qualitative examples in Fig. 4. We compare our model (abbr: Ours) with other baselines: 1) Radial, 2) generator without DH-GAN (abbr: Generator). Specifically, we visualize the visual hints (Note that we add the raw visual hints without pruning compared with the case study in the paper for further illustration): 1) generated by rule-based matching without pruning (abbr: Raw visual hints w/o. pruning), 2) generated with pre-processing visual hints (abbr: Raw visual hints, m=4), 3) predicted with only Generator (abbr: Generator), and 4) predicted with full DH-GAN (abbr: Ours). We can find that our model generates more precise and vivid questions as well as visual hints. Specifically, we find that the raw visual hints without pruning are quite noisy (especially in cases b and c), which fail to guide the question generation procedure faithfully. And the pruned raw visual hints are more referential.

# J   The Details of Error Analysis

See Fig. 5 for error cases of our results. We present one example of each error reason.

**a) Visual hints prediction error.**   It means our model predicts the visual hints incorrectly, which misleads the question generation procedure. The answer "girl" refers to the child holding by the man, but the model misses the correct region representing the "girl". Actually, the model predicts the visual hints representing other men and asks the wrong question.

**b) Detection error.**   It means our model recognizes the objects in the image by mistake. In the image, the woman is riding the tricycle. However, our model recognizes it as "bike" incorrectly.

**c) Reasoning error.**   It means that our model makes wrong logical reasoning. In the picture, the apple is actually ripe. However, the expected answer is "no".

**d) Syntactic error.**   It means our model generates syntactically incorrect questions. The 'What expression ... wearing' is incorrect.


Figure 4: The details of case study examples. The blue rectangles mean the visual-hint regions.

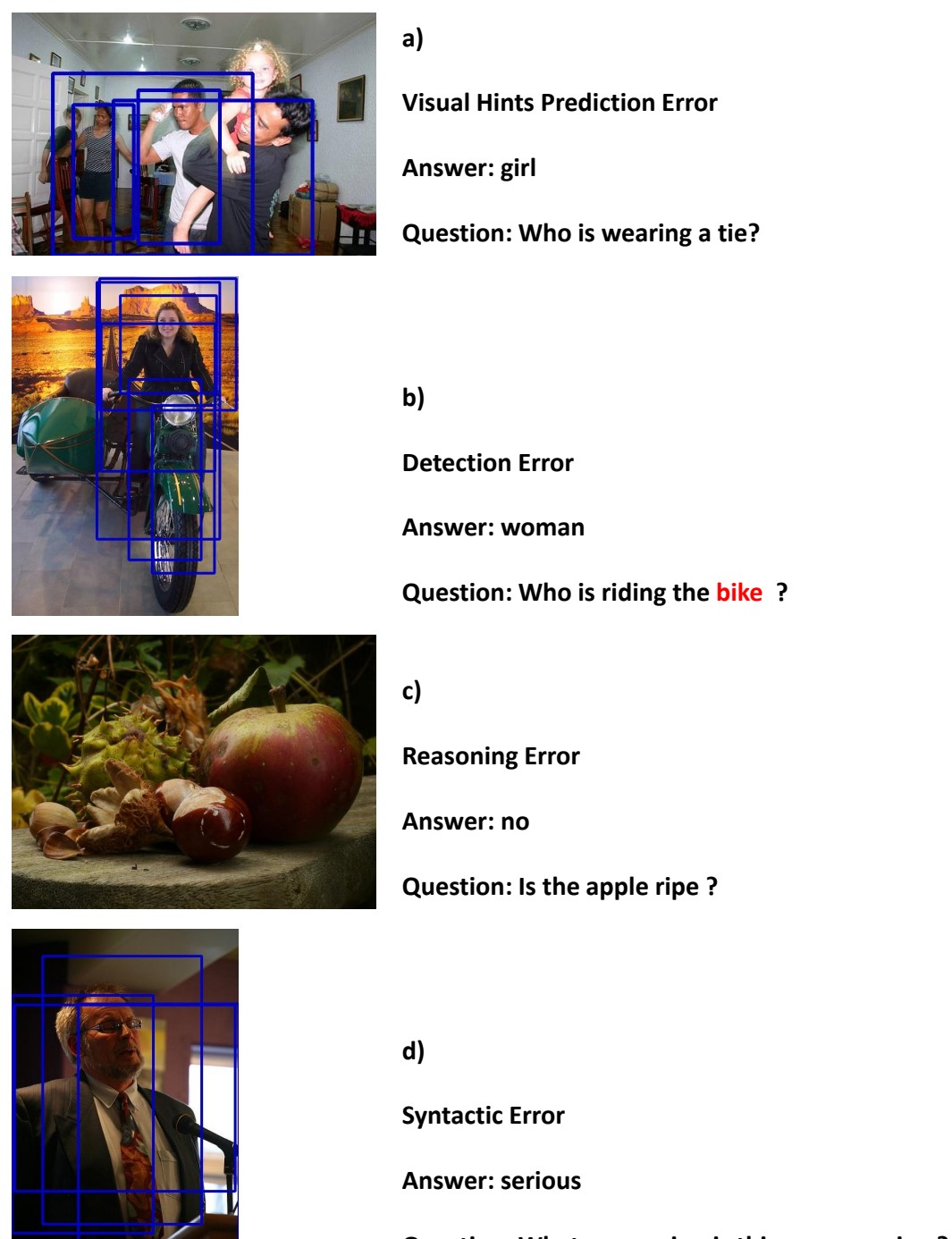

a)

Visual Hints Prediction Error

Answer: girl

Question: Who is wearing a tie?

b)

Detection Error

Answer: woman

Question: Who is riding the **bike** ?

c)

Reasoning Error

Answer: no

Question: Is the apple ripe ?

d)

Syntactic Error

Answer: serious

Question: What expression is this man wearing ?

Figure 5: The details of error examples. The blue rectangles are the predicted visual hints by our model.