# OpenReview forum: "Learning to Generate Visual Questions with Noisy Supervision"
_NeurIPS.cc/2021/Conference — NeurIPS 2021 Poster_

### Official Review · Reviewer_tJc4 · 2021-06-28

**Rating:** 5
**Confidence:** 4

**Summary:**

This work aims to generate human-like visual questions from an image, the questions and answers in training data, and the visual hints via an ad-hoc method using an object detector where detected visual regions and predicted attributes are available following Kai et al., 2021. The improvement over the previous work, DH-VQG, is to adopt GAN-based training to overcome the noisy supervision of the visual hints. Unfortunately, there are weak evidences for the usefulness of the work and it needs to improve the exposition of the work to be self-contained.

**Ethical Concerns:**

None.

**Limitations And Societal Impact:**

The major limitations come from the evaluation to prove the usefulness of the proposed approach. Please clarify the practicality of data augmentation.

**Main Review:**

1) Is it really useful for data augmentation?

1a) As the authors pointed out in L23, the VQG task is “one of the effective data augmentation approaches,” however, in Fig. 3, they showed a minor improvement (<0.2 accuracies around 50% percent of training data) over the baseline. Notice that, for the visual hints, they exploited Stanford CoreNLP (E.1.1) and Faster-RCNN pre-trained on Visual Genome (E.1.3) using its heavy annotations. By the way, one can exploit the visual question answering annotations of Visual Genome for data augmentation to improve the accuracy instead of a hassle of VQG (although the two methods can interfere with each other.)

1b) The automatic evaluation metrics, i.e., BLEU, do not guarantee the usefulness of the generated questions. As the authors pointed out in L206, it measures “the syntactic quality of the generated questions” and “it is well known that the BLEU score cannot faithfully reflect the semantic quality.” Additionally, Table 2 shows that, presumably, the indifference (Could you clarify the confidence interval? What are the numbers in the parentheses?) of the proposed method in the aspect of syntax in the human evaluation. Therefore, Table 1 is supportive evidence, not the ground for the effective evaluation of generated questions.

1c) The BLEU and the human evaluation rely on the ground-truth questions. However, does it promote the usefulness of the generated questions for data augmentation if the generated questions are similar to the ground-truth questions? The proposed method only generated similar expressions of questions sharing the same answer. They do not evaluate the diversity of questions although the authors argue that “to perform both exploration of question sample and the visual hints via policy gradient. (L68)” These issues undermine the purpose of VQG, why VQG is important.

2) Writing and exposition

2a) At the end of Section 2.2.1, the authors do not explain how the target visual hints are obtained. This explanation is critical to understand the proposed method, but the current form is not self-contained. The recommendation is to update a brief mention of the following of Kai et al. (2021) while adding a redirection to E.1.1 and E.1.2 for the details.

3) Minor issues

- The citation of Kai et al., 2021 is missing the name of the conference. (Not limited to this.)
- In L68, “to perform exploration of *both* question samples and visual hints via policy gradient”?
- In L85, how do you define the word tokens?
- In L138, “compute *the* average of relevance score”
- In L149, why do you initialize the starting states by answer embedding?
- In L216, you previously define that $\bar{Q}$ is the sampled (L194) instead of the target question (L216).

----
I want to let you know that we, the reviewers, have a discussion about the concerns raised by the reviewers. The main concerns of the paper are sparse contributions and low significance. Especially, the authors fail to persuade why VQG is an interesting task and what would meaningfully contribute to the community. The connection of zero-shot and few-shot to VQG is weak and it also seems an astray of the main proposition -- weakly supervised learning with VQG.
Therefore, in the revision, I highly recommend strengthening the motivation of VQG and perform rigorous evaluations to show the significance of the proposed idea.

**Time Spent Reviewing:**

5

---

> ### Author Response · Authors · 2021-08-10
> **Authors' response to Reviewer tJc4**
>
> We would like to thank the reviewer for their very constructive comments. The following is our detailed responses regarding all major concerns. We hope the following responses can clarify the missing points and address these concerns.
>
> Q1a and Q1c: Effectiveness of data augmentation from VQG
>
> We fully agree that data augmentation is an important aspect to evaluate the effectiveness of various QG methods besides the automatic metrics and human evaluation, although not every published QG paper has this experiment. However, we would like to point out that generating better questions does not necessarily mean it results in questions similar to the questions samples in existing training dataset. We provide the statistic results about the unseen questions (generated by our method) in total augmented training dataset,
>
> Ratio	|	0.3	|	0.5	|	0.7
>
> Unseen	|	0.3093	|	0.4752	|	0.4432
>
> From the table, we can find that about 30%-47% questions are unseen in the original training split (Ideally, the rate should be 50% if all the generated questions are unseen since we included the original questions). Thus the data augmentation indeed helps improve the VQA performance.
> In Figure 3, our data augmentation experiment shows that when the ratio of original training data is relatively small (0.3), the improvement of the corresponding VQA with generated QA pairs is more significant. This is not surprising since it is common findings across different QG papers that QG is more useful in few-shot or zero-shot settings as demonstrated in recent general QG works [4][5]. In order to further show the potential usefulness of VQG, we followed the experimental settings of VQG/VQA tasks in [6][7] and performed experiments under zero-short settings.
>
> Method           |   BUTD  |    BAN
>
> w/o. VQG.       |   0%.     |.   0%
>
> Radial VQG    |   18.15% | 18.37%
>
> DH-GAN         |  19.21% | 19.93%
>
> As shown in the above table, we can find both Radial and DH-GAN achieve remarkable gains, which demonstrates that the VQG is an effective way to enhance the zero-shot VQA performance, where DH-GAN performs consistently better than Radial VQG.
>
> [4] Liangming Pan, Wenhu Chen, Wenhan Xiong, Min-Yen Kan, William Yang Wang, “Unsupervised Multi-hop Question Answering by Question Generation”, https://arxiv.org/abs/2010.12623.
>
> [5] Siamak Shakeri, Cicero Nogueira dos Santos, Henry Zhu, Patrick Ng, Feng Nan, Zhiguo Wang, Ramesh Nallapati, Bing Xiang, “End-to-End Synthetic Data Generation for Domain Adaptation of Question Answering Systems”, https://arxiv.org/abs/2010.06028.
>
> [6] Xing Xu, Tan Wang, Yang Yang, Alan Hanjalic, and Heng Tao Shen. Radial graph convolutional network for visual question generation. IEEE Transactions on Neural Networks and Learning Systems, 2020.
>
> [7] Y. Li, Y. Yang, J. Wang, and W. Xu, “Zero-shot transfer VQA dataset,” 2018, arXiv:1811.00692. [Online]. Available: http://arxiv. org/abs/1811.00692
>
> Finally, we would like to point out that using pretrained faster-rcnn for preprocessing images and using Stanford CoreNLP for preprocessing texts are all common preprocessing steps in existing papers, which is thought to be lightweight tools for obtaining a better representation of the objects (images or texts), compared to any heavy human annotations. Yes, other possible sources of VQA QA pairs could be used for data augmentation but usually rather limited.
>
> Q1b: Limitations of automatic evaluation metrics
>
> This is a great point. Indeed, for text generation tasks, it is a common agreement in the Generation community that current automatic evaluation metrics like BLEU, ROUGE and so on are not ideal since most of them only measure syntactic correctness (based on n-gram overlap). However, although there are some recent attempts to overcome these limitations, these automatic evaluation metrics still serve as the main evaluation tool for measuring the quality of generated text sequence. Therefore, in order to comprehensively evaluate the results, we have performed various experiments, including general performance on two popular datasets, ablation study, human evaluation, error analysis, and data augmentation to provide enough experimental evidence of how good our DH-GAN method is compared to existing methods.
>
> Q2: More details about target visual hints
>
> This is a great suggestion! We are sorry about the confusion here. The target V is obtained in the preprocessing procedure by similarity matching. Given an image and qa-pair, firstly we use an object detection model (Faster RCNN) to generate a set of objects with labels. Secondly, we adopt Stanford CoreNLP to find the noun phrases. Thirdly, we use GloVE to represent the labels and noun phrases and L2 similarity matching to find the objects(candidate objects) which are similar to the noun phrases. Finally, we use a rule-based pruning method to reduce the number of candidates and the pruned candidates are visual hints V. For more details, please refer to Appendix E.
>
> We will clarify this part in our revision.
>
> Q3: Minor typos issue
>
> Thank you so much for your very careful reading. We will correct them in the revision.

---

> ### Author Response · Authors · 2021-08-30
> **Follow-up Response**
>
> Dear Reviewer tJc4,
>
> Thank you very much for your insightful comments. We have tried our best to clarify and address the concerns and comments by the reviewer in the initial response. We are glad to answer and clarify any further questions and advices from the reviewer for better readability.

---

### Official Review · Reviewer_noBr · 2021-07-16

**Rating:** 4
**Confidence:** 4

**Summary:**

The paper proposes to improve visual question generation by using semi-supervised visual grounding information. However, that is not the main contribution of the paper, as it is proposed by a closely related work [19]. Instead, the paper focuses on how to *better* use the information/hint from noisy visual grounding supervision obtained from visual saliency-based approaches. To this end, the paper proposes to use GAN to indirectly supervise the *goodness* of the generated visual region hint by trying to discern real/fake question, answer, image triplet. Experiments are done on the existing VQG dataset show some improvement over [19] and other baselines using both auto-metrics and human judgments.

**Ethical Concerns:**

The posted link to anonymized github repo shows one of the authors as being contributors. While this knowledge has not affected my impartiality, it could potentially be problematic/used as signal to identify the authors.

**Limitations And Societal Impact:**

There are some concerns about the instructions given, qualifications, and compensation structure for the human annotators. However, I think it is a relatively small-scale experiment and is not of huge concern.

**Main Review:**

**Strengths**

**[S1]** Sensible model design: The design of real/fake GAN to deal with noisy visual grounding obtained from [19] is interesting and it is easy to see why that might help the model learn better.

**[S2]** Clearly written with exception of pervasive grammar errors: There are several errors in grammar and phrasing which does need to be rectified, but they do not always make the text difficult to understand. (E.g., in the abstract: "Despite they proposed a simple rule-based similarity matching method to obtain candidate visual hints, they could be very noisy practically"). The main techniques, motivations, and experimental results are presented clearly and are easy to understand.

**Weaknesses**

**[W2]** Contributions are sparse and of rather low significance: The idea to use visual grounding information (called double hint) is already explored by the closest work [19]. The main contribution is the use of GAN to deal with noisy supervision which is bound to arise due to the rule-based nature of generating those visual hints in [19]. The formulation of GAN for this purpose is also relatively straightforward and furthermore, it is unclear how much better the proposed method is, compared to [19]. Overall, the novelty and significance of this work is quite low.

**[W1]** Human evaluation is not described well and not meaningful: Firstly, the human evaluation results omit the closest work, DH-VQG which is very concerning. Next, there is next to no detail about the exact instructions given to evaluators, what the criteria "syntax, semantics, and relevance" mean (and how they were shown to annotators). There is no justification about why a Likert scale was used instead of head-to-head comparisons etc..We don't even know who the annotators were or how they were evaluated/vetted/compensated. Finally, the final reported numbers are quite close together for baseline and proposed and it is unclear how to interpret that. Overall, the human evaluation adds little to no information due to the numerous problems mentioned above.

**[W3]** VQG is not widely studied and the impact of the work is not clear: VQG, on its own, is not very popular in the community as it remains unclear what its benefits would be (data augmentation?). I know this is not the scope of the paper but the concerns about its significance remain intact. The nearest related work [19] is not published yet (mostly due to low contributions/significance according to open review) and the current work is a slight improvement upon that. As such, I am unsure how important/meaningful the contributions are. The techniques proposed are also unlikely to scale to anything other than VQG.

**Overall:**
Overall, I think the paper is a good effort and shows promising results. However, the proposed method is only marginally different/better than its nearest related work. Furthermore, the proposed method is likely to be limited in application to VQG, a problem that has not clearly established itself in the field. There are also minor complaints about the reporting of the results and human evaluation. I recommend rejection of this work in its current form.


**Time Spent Reviewing:**

3

---

> ### Author Response · Authors · 2021-08-10
> **Authors' response to Reviewer noBr**
>
> We would like to thank the reviewer for their very critical but thoughtful comments. The following is our detailed responses regarding all major concerns. We hope the following responses can clarify the missing points and address these concerns.
>
> Q1:  Novelty concerns: 1) differences compared with [19]; 2) bounded performance improvement
>
> We are sorry that there are some misunderstandings regarding what’s new in our paper. We hope the following responses can clarify these missing points.
>
> 1. how to compared with [19]
>
> Connection with [19]. As discussed in lines 47-60 in Introduction, our paper followed the learning settings - double visual and textual hints for VQG first proposed in [19], which has proven to be state-of-the-art for VQG compared against other three learning settings (without any hints, with only answer types, and with answer hints). The Key Observation is that visual hints are not available naturally and a set of visual hints obtained using a simple rule-based similarity matching method in [19] are often too noisy for generating high-quality questions. This motivated us to develop an effective approach for identifying salient visual regions of interest under the noisy supervision in order to address the Question we asked in lines 53-55.
>
> Differences with [19]. First of all, the most important difference between our method DH-GAN and the method DH-Graph2Seq in [19] is how to identify salient visual hints without any ground-truth signals (human annotations). As we mentioned above, [19] used a simple rule-based similarity matching method. In contrast, we cast this problem as a weakly supervised learning problem with noises, where we focus on estimating the probability of sample triplet ( i.e., the image, answer, and question) in order to implicitly measure the quality of predicted salient visual hints. To achieve this, we introduced a double-hints based question generator and a question-answer-aware discriminator. Second, the main model architecture of DH-Graph2Seq [19] is based on Graph2Seq learning paradigm while our method DH-GAN is based on generator-evaluator learning paradigm, which is fundamentally different from each other. Third, we also employed RL with a novel design of reward functions suitable for VQG tasks while [19] did not have such a component.
>
> In summary, our method is built on the learning setting first proposed in [19] and we propose a novel learning approach under noisy supervision for addressing important limitations in [19], which outperforms the state-of-the-art approaches by a large margin on a variety of metrics, including both automatic machine metrics and human evaluation.
>
> 2. bounded performance improvement
>
> We respectfully disagree with this argument. Looking back the history of development of VQG (and other QG related tasks) [1][2][3], there are roughly divided into four learning settings and stages: i) VQG without any hints, ii) VQG with answer type hint; iii) VQG with textual answer hints; iv) VQG with double visual and answer hints, as we clearly discussed in the Introduction section. Among each of these learning settings and stages, there are different learning approaches developed for continuously improving the performance of VQG.
>
> Our paper is built on a recent state-of-the-art method [19] with a new learning setting - VQG with double visual and answer hints. Our paper is completely motivated by the noisy visual hints in [19] and proposed a principled approach for elegantly addressing this issue by introducing a noisy and weak supervision to implicitly measure the quality of predicted salient visual regions of interest. Motivated by this learning rationale, we developed a generator-evaluator model architecture for achieving this. It is NOT the other way around that we simply used GAN for our application. Besides, the way on how to design our generator, discriminator, as well as RL rewards are NOT trivial without showing how to do them in this paper. Our extensive experiments on general performance on two popular datasets, ablation study, human evaluation, error analysis, and data augmentation are the experimental evidence of how good our DH-GAN method is compared to existing methods. We do agree that it is better to add DH-Graph2Seq [19] (their codes are not publicly released yet)  in human evaluation and data augmentation (which we will add in our revision thanks to the reviewer’s comments).
>
> [1] Liangming Pan, Wenqiang Lei, Tat-Seng Chua, Min-Yen Kan, “Recent Advances in Neural Question Generation”, 2019.
>
> [2] Ghader Kurdi, Jared Leo, Bijan Parsia, Uli Sattler, Salam Al-Emari, “A Systematic Review of Automatic Question Generation for Educational Purposes”, 2020.
>
> [3] https://github.com/teacherpeterpan/Question-Generation-Paper-List
>
> Q2: The details of human evaluation; missing baseline [19]
>
> We are sorry for the confusion. Following previous papers in the literature, we omit the very detailed information regarding the experimental setting in human evaluation and refer the readers to the detailed discussion in Appendix H, as discussed in Line 272. We would like to add [19] but unfortunately we did not see any publicly released codes from [19] . We will try to make them more clear in our revision.
>
> Q3: The impact of VQG and impact of proposed method
>
> Visual question generation task is just like other question generation tasks, which has been well defined, studied and still growing research subdomain in recent years [1][2][3]. For all QG tasks (beyond visual QG), as long as the corresponding QA and VQA tasks are still data hungry, we could not see a reason why its dual tasks QG and VQG are not important to the communities. Recent progresses in applying QG in domain adaptation or transfer learning settings [4][5][6][7] are a new attempt for making full use of QG tasks.
>
>  [4] Liangming Pan, Wenhu Chen, Wenhan Xiong, Min-Yen Kan, William Yang Wang, “Unsupervised Multi-hop Question Answering by Question Generation”, https://arxiv.org/abs/2010.12623.
>
> [5] Siamak Shakeri, Cicero Nogueira dos Santos, Henry Zhu, Patrick Ng, Feng Nan, Zhiguo Wang, Ramesh Nallapati, Bing Xiang, “End-to-End Synthetic Data Generation for Domain Adaptation of Question Answering Systems”, https://arxiv.org/abs/2010.06028.
>
> [6] Xing Xu, Tan Wang, Yang Yang, Alan Hanjalic, and Heng Tao Shen. Radial graph convolutional network for visual question generation. IEEE Transactions on Neural Networks and Learning Systems, 2020.
>
> [7] Y. Li, Y. Yang, J. Wang, and W. Xu, “Zero-shot transfer VQA dataset,” 2018, arXiv:1811.00692. [Online]. Available: http://arxiv. org/abs/1811.00692
>
> Regarding the applicability of our proposed method, we deeply believe that our proposed learning rationale of this paper could be widely applicable in other similar learning/applications tasks. The key learning rationale is to cast a secondary task without ground-truth (predicting the salient visual region of interests in this paper) as a constraint to the main task (visual question generation in this paper), and to use an evaluator to implicitly measure the goodness of the prediction of this secondary task. It would be particularly useful for other vision and language tasks.

---

> ### Author Response · Authors · 2021-08-30
> **Follow-up Response**
>
> Dear Reviewer noBr,
>
> Thank you very much for your insightful comments. We have tried our best to clarify and address the concerns and comments by the reviewer in the initial response. We are glad to answer and clarify any further questions and advices from the reviewer for better readability.

---

### Official Review · Reviewer_W4CH · 2021-07-17

**Rating:** 5
**Confidence:** 4

**Summary:**

The work proposes a double hint guided GAN architecture, consisting of a visual and answer hint based generator and question+answer+image triplet discriminator. The work also proposes a policy gradient based reward function to optimize both question generation and visual hint prediction. The proposed method outperforms baselines on several metrics and human evaluation studies.

**Limitations And Societal Impact:**

Authors have not added any discussion about societal impact of the work.

**Main Review:**

STRENGTHS
- The GAN based VQG architecture achieves better generation result compared to DH-VQG and other baselines on two datasets VQA and COCO-QA.
- Ablation studies were performed on the different hyper-parameters used for the setup.


WEAKNESSES AND REMARKS

- Does VQG task generated questions actually help in improving VQA performance? The premise that using answer hints and visual features help to generate better questions can also result in questions which are very similar to already existing questions in the dataset. Hence addition of such new questions might not improve the VQA task performance significantly. Experiments in Section 3.4 Data Augmentation, it is unclear how any additional questions are added during data augmentation for Radial and DH-GAN results. Interestingly as more questions from the train set are added, the benefits of VQG starts to diminish and all the baselines and DH-GAN starts to perform very same. To some extent that justifies the argument that the generated questions might be just very similar questions that are already in the training set.

- To understand more about the quality of the generated questions, data augmentation experiment is very important. For fair comparison, I would be interested to see how another dual vision+answer hints approaches like DH-VQG work compared to DH-GAN approach. Since the major difference between this work and DH-VQG is the difference in architecture, it is important to see whether the improvement is coming from using visual hints or the GAN based approach.

- Addition of dual visual features and answer hints has already been proposed by DH-VQG[19] paper. This paper applies the same idea but uses a different architecture for question generation. I am unsure how much additional novelty this approach adds.

- How does $k$ (number of modules in the visual hint generator) effect the quality of the visual hints and the eventual generations? This ablation is missing.

- The model architecture is too complex to understand. Latest state-of-the-art multimodal models are using transformer layers to learn attention between different modalities like image features and text, and have been shown to perform better than LSTMs. I would like to understand from the authors if they considered using transformers to learn attention between visual features and answers to generate visual hints over using multiple LSTMs int he generator, that would simplify the architecture.

- Human evaluation is missing DH-VQG[19] baseline, which seems to be the most closely related approach. In addition, human evaluation results are very close to that of only Generator baseline.

I would be interested to see the discussion and authors' replies to the concerns during the response stage in order to consider improving the rating.

**Time Spent Reviewing:**

5

---

> ### Author Response · Authors · 2021-08-10
> **Authors' response to Reviewer W4CH**
>
> We would like to first thank the reviewer’s critical and constructive comments. The following is our detailed responses regarding all major concerns. We hope the following responses can clarify the missing points and address these concerns.
>
> Q1: Effectiveness of data augmentation from VQG
>
> We fully agree that data augmentation is an important aspect to evaluate the effectiveness of various QG methods besides the automatic metrics and human evaluation, although not every published QG paper has this experiment. However, we would like to point out that generating better questions does not necessarily mean it results in questions similar to the questions samples in existing training dataset. We provide the statistic results about the unseen questions (generated by our method) in total augmented training dataset,
>
> Ratio	|	0.3	|	0.5	|	0.7
>
> Unseen	|	0.3093	|	0.4752	|	0.4432
>
> From the table, we can find that about 30%-47% questions are unseen in the original training split (Ideally, the rate should be 50% if all the generated questions are unseen since we included the original questions). Thus the data augmentation indeed helps improve the VQA performance.
> In Figure 3, our data augmentation experiment shows that when the ratio of original training data is relatively small (0.3), the improvement of the corresponding VQA with generated QA pairs is more significant. This is not surprising since it is common findings across different QG papers that QG is more useful in few-shot or zero-shot settings as demonstrated in recent general QG works [4][5]. In order to further show the potential usefulness of VQG, we followed the experimental settings of VQG/VQA tasks in [6][7] and performed experiments under zero-short settings.
>
> Method           |   BUTD  |    BAN
>
> w/o. VQG.       |   0%.     |.   0%
>
> Radial VQG    |   18.15% | 18.37%
>
> DH-GAN         |  19.21% | 19.93%
>
> As shown in the above table, we can find both Radial and DH-GAN achieve remarkable gains, which demonstrates that the VQG is an effective way to enhance the zero-shot VQA performance, where DH-GAN performs consistently better than Radial VQG.
>
> [4] Liangming Pan, Wenhu Chen, Wenhan Xiong, Min-Yen Kan, William Yang Wang, “Unsupervised Multi-hop Question Answering by Question Generation”, https://arxiv.org/abs/2010.12623.
>
> [5] Siamak Shakeri, Cicero Nogueira dos Santos, Henry Zhu, Patrick Ng, Feng Nan, Zhiguo Wang, Ramesh Nallapati, Bing Xiang, “End-to-End Synthetic Data Generation for Domain Adaptation of Question Answering Systems”, https://arxiv.org/abs/2010.06028.
>
> [6] Xing Xu, Tan Wang, Yang Yang, Alan Hanjalic, and Heng Tao Shen. Radial graph convolutional network for visual question generation. IEEE Transactions on Neural Networks and Learning Systems, 2020.
>
> [7] Y. Li, Y. Yang, J. Wang, and W. Xu, “Zero-shot transfer VQA dataset,” 2018, arXiv:1811.00692. [Online]. Available: http://arxiv. org/abs/1811.00692
>
> Q2: Comparison of DH-GAN with DH-Graph2Seq [19] (or DH-VQG) in data augmentation experiment
>
> This is a very fair request but we don’t have DH-Graph2Seq code (since they did not release it). Due to limited time, we are not able to add this experiment. We will add this experiment in our revision. However, we would like to point out that In Table 2 and 3, Generator is the model that can be roughly treated to be similar to DH-Graph2Seq [19] although the specific model architecture is different. We hope this could give you some reference points to compare.
>
> Q3: Novelty concerns: differences compared with [19]
>
> We are sorry that there are some misunderstandings regarding what’s new in our paper. We hope the following responses can clarify the missing points and address these concerns.
>
> Connection with [19]. As discussed in lines 47-60 in Introduction, our paper followed the learning settings - double visual and textual hints for VQG first proposed in [19], which has proven to be state-of-the-art for VQG compared against other three learning settings (without any hints, with only answer types, and with answer hints). The Key Observation is that visual hints are not available naturally and a set of visual hints obtained using a simple rule-based similarity matching method in [19] are often too noisy for generating high-quality questions. This motivated us to develop an effective approach for identifying salient visual regions of interest under the noisy supervision in order to address the Question we asked in lines 53-55.
>
> Differences with [19]. First of all, the most important difference between our method DH-GAN and the method DH-Graph2Seq in [19] is how to identify salient visual hints without any ground-truth signals (human annotations). As we mentioned above, [19] used a simple rule-based similarity matching method. In contrast, we cast this problem as a weakly supervised learning problem with noises, where we focus on estimating the probability of sample triplet ( i.e., the image, answer, and question) in order to implicitly measure the quality of predicted salient visual hints. To achieve this, we introduced a double-hints based question generator and a question-answer-aware discriminator. Second, the main model architecture of DH-Graph2Seq [19] is based on Graph2Seq learning paradigm while our method DH-GAN is based on generator-evaluator learning paradigm, which is fundamentally different from each other. Third, we also employed RL with a novel design of reward functions suitable for VQG tasks while [19] did not have this component.
>
> In summary, our method is built on the learning setting first proposed in [19] and we propose a novel learning approach under noisy supervision for addressing important limitations in [19], which outperforms the state-of-the-art approaches by a large margin on a variety of metrics, including both automatic machine metrics and human evaluation.
>
> Q4: How does  (number of modules in the visual hint generator) effect the quality of the visual hints and the eventual generations?
>
> This is a great suggestion. We have performed a new experiment investigating the effect of K on the final evaluation performance (on BLEU-4). As shown in the following table, our approach is not sensitive to the hyperparameter K.
>
> Number of k	|	BLEU@4
>
> k=1	|	23.42
>
> k=2	|	23.60
>
> k=3	|	23.71
>
> k=4	|	23.68
>
> k=5	|	23.44
>
> Q5: Transformer layers instead of LSTMs for VQG
>
> Yes, this is a great suggestion. Indeed, we also noticed these new research results on Transformers on multimodality data. We also have considered using transformer architecture and reached similar results when compared with the LSTM based architecture in the pretraining-stage. However, we would like to point out that the transformer is slower in the adversarial training stage since we have to generate the whole questions in each step autoregressively. In addition, the transformer is more data hungry compared to LSTM based architectures.
>
> Q6: Human evaluation is missing DH-VQG[19] baseline; human evaluation results are very close to that of only Generator baseline.
>
> We would like to but unfortunately we did not see any publicly released codes from [19] . Regarding human evaluation In Table 2, although DH-GAN has similar syntax and semantics scores in human evaluation, DH-GAN has higher Relevance score than Generator, highlighting the importance of identifying correct salient visual hints.

---

> ### Author Response · Authors · 2021-08-30
> **Follow-up Response**
>
> Dear Reviewer W4CH,
>
> Thank you very much for your insightful comments. We have tried our best to clarify and address the concerns and comments by the reviewer in the initial response. We are glad to answer and clarify any further questions and advices from the reviewer for better readability.

---

> ### Author Response · Authors · 2021-08-31
> **Post-rebuttal Comments**
>
> Thanks again for taking the time to process and review our submission!
>
> We are wondering if our rebuttals have addressed your previous concerns. If you have any further concerns or comments, please feel free to ask. We would be more than happy to provide our responses.

---

### Official Review · Reviewer_6CYj · 2021-07-19

**Rating:** 5
**Confidence:** 4

**Summary:**

In this paper, the authors proposed a new visual question generation method called Double-hint GAN (DH-GAN). In this method, the authors leveraged both the visual hints and the answer to generate the corresponding question. To make it grounding on the specific visual and answer hints, the authors introduced supervised losses to guide the pointing of visual objects and question decoding. Besides, the authors used a discriminator to learn to differentiate the ground-truth question and generated question, and in turn, the output of the discriminator is used as the reward along with the auto metric BLEU. After training with a fusion of supervised learning and policy gradient REINFORCE, the model demonstrates superior performance on both VQA 2.0 and COCO-QA to previous works.

The main contribution of this paper is that the authors proposed to use a pipeline similar to the generative adversarial network (GAN) for visual question generation. In the paper, a discriminator was introduced to impose the reward for policy gradient besides the regular supervised training loss and the reward from the automatic metric BLEU. The experimental results show that this method can generate better questions conditioned on the given visual hints and target answers, compared with the prior works.

**Limitations And Societal Impact:**

The authors did not discuss the limitations and societal impact. But again, I think the main concern is that the paper lacks novelty and the task of visual question generation stand-alone is ill-posed without the showcase of its notable benefit to downstream tasks, such as VQA.

**Main Review:**

[Originality]

This paper lacks novelty from various perspectives. First, the proposed method is similar to [19] in that they both used similar so-called double hints to generate the questions, which include visual hints and answers. Moreover, the visual hint pointing loss and the question decoding loss used in this paper are also similar. Second, different from [19], this paper used a REINFORCE method to guide the question generation. More specifically, the authors used two rewards, one is from the automatic metric BLEU, and the other one is from the discriminator trained during the whole learning process. Obviously, using the automatic metric as the reward has been used for image captioning [5,38], and the discriminator here is a kind of ad-hoc component to the main pipeline.

[Quality]

Pros:

1. This paper proposed a new pipeline called DH-GAN for visual question generation. It consists of two main components, a generator that takes visual hints and answers for question generation, and a discriminator learning to distinguish the ground-truth answer and the generated answer given the image, visual hints, and the answer.

2. The proposed DH-GAN demonstrates superior performance to previous work including DH-VQG on two datasets. Further ablation studies demonstrate that visual hints are important to guide question generation.

3. The authors further tried to use the generated questions to perform data augmentation for visual question answering. This is an interesting and meaningful study.

Cons:

1. The visual question generation is somehow an ill-posed problem. Given the image and answer, there are multiple questions that can lead to the same answer. Even though this paper further leveraged the visual hints to reduce the size of candidate space, it is still an ambiguous problem. Forcing the model to predict the correct question is more like an overfitting game.

2. As discussed above, the method proposed in this paper does not have enough novelty from different perspectives. Particularly, this paper is very similar to [19] in terms of the objective function and model pipeline. Though the authors introduced a new discriminator, there is no clear cue about how this discriminator contributes to the final performance, not to say the discriminator is also not new.

3. The authors claimed that the proposed method outperforms the previous method by a large margin. However, to me, it is close to DH-VQA on various metrics in Table 1, which makes sense because they both used visual hints, i.e., detected objects from the image.

4. The ablation study in this paper conveys little information. According to Table 3, the main impression is that the visual hints are important to generate good questions. However, using visual hints is not the novel part of this paper. It would be better to have more ablation studies on how the discriminator helps to question generation.

5. In Sec 3.4, the authors used generated questions for data augmentation. As we can see, the proposed model outperforms the baseline method very marginally, which weakens the contribution of this paper. Note that the model is not aimed at improving VQA during the training. Instead, it always learns to generate the questions which have been already covered in the original dataset. As such, it is more like a heavily conditioned and ill-posed image captioning problem.

[Clarity]

1. The difference between the proposed method and [19] is not clearly discussed. In lines 50-52, the authors mentioned [19] used a simple rule-based method to obtain the visual hints. I do not think this is a correct statement because [19] also used a learnable module to learn to point to the target visual hints. Besides [19], the authors also did not discuss the difference to other methods in the related work section.

2. Many details are missed in the paper. For example, how many objects are extracted from the image? What is the training procedure for the proposed pipeline? How often the discriminator is updated during the training? Is the REINFORCE added at the beginning of the training or later stage? How many visual hints predictor modules are used in the paper and how does it affect the final performance? How to compute the reward D(\cdot) in Eq. (12)?

3. In Table 1, the authors should add cites for previous works; "double text and visual hints" is a weird and annoying term. I do not quite get the usage of "+" and "-" in Table 3. The different error types shown in Figure 3 are not explained.

[Significance]

This paper proposed a new method for visual question generation, which demonstrates better performance on VQA 2.0 and COCO-QA. As discussed above, the novelty of this paper is limited. Also, the motivation of this paper is not clear, and the contribution of this paper is not well-justified in terms of the improvements and the benefit to downstream task VQA. Therefore, I think this paper is not ready for this venue. I would suggest the authors focus more on how to leverage the visual question generation for visual question answering or other end tasks.

3.

**Time Spent Reviewing:**

16

---

> ### Author Response · Authors · 2021-08-10
> **Authors' response to Reviewer 6CYJ**
>
> We would like to first thank you reviewer for your valuable and constructive comments. The following is our detailed responses regarding all major concerns.
>
> Q1:  Novelty concerns: 1) differences compared with [19]; 2) reward design differences compared to existing works; 3) discriminator here is a kind of ad-hoc component to the main pipeline.
>
> We are sorry that there are some misunderstandings regarding what’s new in our paper. We hope the following responses can clarify the missing points and address these concerns.
>
> 1) differences compared with [19]
>
> Connection with [19]. As discussed in lines 47-60 in Introduction, our paper followed the learning settings - double visual and textual hints for VQG first proposed in [19], which has proven to be state-of-the-art for VQG compared against other three learning settings (without any hints, with only answer types, and with answer hints). The Key Observation is that visual hints are not available naturally and a set of visual hints obtained using a simple rule-based similarity matching method in [19] are often too noisy for generating high-quality questions. This motivated us to develop an effective approach for identifying salient visual regions of interest under the noisy supervision in order to address the Question we asked in lines 53-55.
>
> Differences with [19]. First of all, the most important difference between our method DH-GAN and the method DH-Graph2Seq in [19] is how to identify salient visual hints without any ground-truth signals (human annotations). As we mentioned above, [19] used a simple rule-based similarity matching method. In contrast, we cast this problem as a weakly supervised learning problem with noises, where we focus on estimating the probability of sample triplet ( i.e., the image, answer, and question) in order to implicitly measure the quality of predicted salient visual hints. To achieve this, we introduced a double-hints based question generator and a question-answer-aware discriminator. Second, the main model architecture of DH-Graph2Seq [19] is based on Graph2Seq learning paradigm while our method DH-GAN is based on generator-evaluator learning paradigm, which is fundamentally different from each other. Third, we also employed RL with a novel design of reward functions suitable for VQG tasks while [19] did not have such a component.
>
> In summary, our method is built on the learning setting first proposed in [19] and we propose a novel learning approach under noisy supervision for addressing important limitations in [19], which outperforms the state-of-the-art approaches by a large margin on a variety of metrics, including both automatic machine metrics and human evaluation.
>
> 2) reward design differences compared to existing works
>
> Yes, BLEU score is a well-adopted reward function, which we did not claim that it is novel. Indeed, BLEU score is good at measuring the syntactic quality of text but not the semantic quality of text. Therefore, we proposed to select the discriminator’s probability output as a reward to measure the semantic quality, which combines with the BLEU score to formulate our novel hybrid reward function in Eq. (12).
>
> 3) discriminator here is a kind of ad-hoc component to the main pipeline.
>
> As we discussed in 1), without any ground-truth signals (human annotations), we implicitly measure the quality of predicted salient visual hints through estimating the probability of sample triplet ( i.e., the image, answer, and question), which is the main role of our discriminator. This is an important and principled design to the main pipeline.
>
> Q2: The visual question generation is somehow an ill-posed problem...Forcing the model to predict the correct question is more like an overfitting game.
> We respectfully disagree. Visual question generation task is just like other question generation tasks, which has been well defined, studied and still growing research subdomain in recent years [1][2][3]. For all QG tasks (beyond visual QG), one of the fundamental research challenges is to overcome one-to-many mapping issues.
>
> [1] Liangming Pan, Wenqiang Lei, Tat-Seng Chua, Min-Yen Kan, “Recent Advances in Neural Question Generation”, 2019.
>
> [2] Ghader Kurdi, Jared Leo, Bijan Parsia, Uli Sattler, Salam Al-Emari, “A Systematic Review of Automatic Question Generation for Educational Purposes”, 2020.
>
> [3] https://github.com/teacherpeterpan/Question-Generation-Paper-List
>
> Q3: Novelty concerns: 1) differences compared with [19]; 2) role of discriminator, contribution of discriminator
>
> We refer the reviewer to the detailed responses in Q1. Regarding the performance contribution of the discriminator component, we have shown its effectiveness and importance in our ablation study experiment as shown in Table 3 and Table 2. DH-GAN is our model and Generator is the one we removed the discriminator component. We can see that DH-GAN has about 1.1 higher BLEU-4 (in absolute value) compared to Generator only, which should be treated as significantly higher with such BLEU score difference in Table 3. In Table 2, although DH-GAN has similar syntax and semantics scores in human evaluation, DH-GAN has higher Relevance score than Generator, highlighting the importance of identifying correct salient visual hints.
>
> Q4: Performance differences in Table 1.
>
> In Table 1, our method DH-GAN has about 1.3 higher BLEU-4 and 1.1 higher ROUGE than DH-VQG on dataset VQG2.0, and about 0.4 higher BLEU-4 and 0.8 higher ROUGE than DH-VQG, respectively, not to mention all other scores like CIDRr, METEOR, and SPICE. This is indeed a significant improvement when researchers looked at these score differences. When we further look at the performance differences between all other previous methods listed in Table 1, we believe our claim “outperforms the state-of-the-art approaches by a large margin on a variety of metrics” is fair and appropriate.
>
> Q5: Suggestions on ablation study.
>
> We appreciate the reviewer’s comments. Yes, indeed, we have shown the importance of discriminator by comparing DH-GAN (full model) with Generator. In addition, visual hints together with answer hints (double hints) are definitely a very important factor for final performance, which is the exact motivation why we are developing an effective approach for identifying salient visual regions of interest under the noisy supervision.
>
> Q6: Effectiveness of data augmentation from VQG
>
> We fully agree that data augmentation is an important aspect to evaluate the effectiveness of various QG methods besides the automatic metrics and human evaluation, although not every published QG paper has this experiment. In Figure 3, our data augmentation experiment shows that when the ratio of original training data is relatively small (0.3), the improvement of the corresponding VQA with generated QA pairs is more significant. This is not surprising since it is common findings across different QG papers that QG is more useful in few-shot or zero-shot settings as demonstrated in recent general QG works [4][5]. In order to further show the potential usefulness of VQG, we followed the experimental settings of VQG/VQA tasks in [6][7] and performed experiments under zero-short settings.
>
> Method           |   BUTD  |    BAN
>
> w/o. VQG.       |   0%.     |.   0%
>
> Radial VQG    |   18.15% | 18.37%
>
> DH-GAN         |  19.21% | 19.93%
>
> As shown in the above table, we can find both Radial and DH-GAN achieve remarkable gains, which demonstrates that the VQG is an effective way to enhance the zero-shot VQA performance, where DH-GAN performs consistently better than Radial VQG.
>
> [4] Liangming Pan, Wenhu Chen, Wenhan Xiong, Min-Yen Kan, William Yang Wang, “Unsupervised Multi-hop Question Answering by Question Generation”, https://arxiv.org/abs/2010.12623.
>
> [5] Siamak Shakeri, Cicero Nogueira dos Santos, Henry Zhu, Patrick Ng, Feng Nan, Zhiguo Wang, Ramesh Nallapati, Bing Xiang, “End-to-End Synthetic Data Generation for Domain Adaptation of Question Answering Systems”, https://arxiv.org/abs/2010.06028.
>
> [6] Xing Xu, Tan Wang, Yang Yang, Alan Hanjalic, and Heng Tao Shen. Radial graph convolutional network for visual question generation. IEEE Transactions on Neural Networks and Learning Systems, 2020.
>
> [7] Y. Li, Y. Yang, J. Wang, and W. Xu, “Zero-shot transfer VQA dataset,” 2018, arXiv:1811.00692. [Online]. Available: http://arxiv. org/abs/1811.00692
>
> Q7: More discussions of the difference to other methods in Related Work section
>
> Thanks for the great suggestion. We will add more discussions in our revision when we have one additional page.
>
> Q8: More details regarding experimental settings.
>
> Thanks for the great suggestion. We extract 36 objects from the image (Appendix E.1.3). The training details are in Appendix B. We use 3 visual hints predictor modules (Appendix E.2).
> The D(x) is the (sigmoid) output of the discriminator. We have performed a new experiment investigating the effect of the number of visual hints prediction layers (K) on the final evaluation performance (on BLEU-4). As shown in the following table, our approach is not sensitive to the hyperparameter K.
>
> Number of k	|	BLEU@4
>
> k=1	|	23.42
>
> k=2	|	23.60
>
> k=3	|	23.71
>
> k=4	|	23.68
>
> k=5	|	23.44
>
>
> Q9: Baseline citations in Table 1, term confusion, the purpose of “+” and “-” in Table 3, and error types in Figure 3.
>
> We apologize for the confusion here and appreciate your suggestions. Baselines methods in Table 1 are discussed in Lines 237-246 and cited there. The sign “+” means having this component and “-” means removing this component. We will add more explanations about error types in Figure 3.

---

> ### Author Response · Authors · 2021-08-30
> **Follow-up response**
>
> Dear Reviewer 6CYj,
>
> Thank you very much for your insightful comments. We have tried our best to clarify and address the concerns and comments by the reviewer in the initial response. We are glad to answer and clarify any further questions and advices from the reviewer for better readability.

---

### Decision · Program_Chairs · 2021-09-28

**Decision:**

Accept (Poster)

**Comment:**

All reviewers recommend reject after reviewing the author response and discussion.

While the author response addressed some concerns, several critical concerns remained, including:
1) Limited novelty over prior work.
2) Limited focus on the benefit of VQG, although the authors acknowledge the importance in Section 3.4 and the author response.
2a) Section 3.4. misses comparison to closes competitor DH-VQG and ablations.
2b) The additional results provided in the author response on few / zero-shot are interesting, but again miss comparison to prior work (e.g. DH-VQG) and ablations for this important final task

I thus recommend reject.
[The authors might want to consider submitting a significantly revised version to a vision or NLP focused venue]

**Consistency Experiment:**

NeurIPS has a long history of experimentation. In 2014, NeurIPS ran an experiment in which 10% of submissions were reviewed by two independent committees to quantify the randomness in the review process. This year, we repeated a variant of this experiment to see how the quality of the review process has changed over time.  This paper was part of the experiment and was therefore assigned to two committees (consisting of reviewers, an Area Chair, and a Senior Area Chair) that reached independent decisions.  If both committees made the same recommendation, this recommendation was followed. If a single committee recommended acceptance, the paper was accepted (with the exception of a few cases in which the other committee identified what we considered a fatal flaw, e.g., an error in a key result).

This copy’s committee reached the following decision: **Reject**

The other committee assigned to the paper recommended **Accept (Spotlight)**.  You can find the other set of reviews, along with any follow up discussion with the authors here:
https://openreview.net/forum?id=gEGPcdiXbky